# Unlocking Visual Secrets: Inverting Features with Diffusion Priors for Image Reconstruction

**Sai Qian Zhang***                                                    *sai.zhang@nyu.edu*

*New York University*

**Ziyun Li**                                                          *liziyun@meta.com*
*Reality Labs Research, Meta*

**Chuan Guo**                                                        *chuanguo@meta.com*
*Fundamental AI Research (FAIR), Meta*

**Saeed Mahloujifar**                                                *saeedm@meta.com*
*Fundamental AI Research (FAIR), Meta*

**Deeksha Dangwal**                                                  *saeedm@meta.com*
*Reality Labs Research, Meta*

**Edward Suh**[†]                                                    *suh@ece.cornell.edu*
*Nvidia/Cornell University*

**Barbara De Salvo**                                                 *barbarads@meta.com*
*Reality Labs Research, Meta*

**Chiao Liu**                                                        *chiaoliu@meta.com*
*Reality Labs Research, Meta*

**Reviewed on OpenReview:** *https://openreview.net/forum?id=j6MgbuBiGV*

## Abstract

Inverting visual representations within deep neural networks (DNNs) presents a challenging and important problem in the field of security and privacy for deep learning. The main goal is to invert the features of an unidentified target image generated by a pre-trained DNN, aiming to reconstruct the original image. Feature inversion holds particular significance in understanding the privacy leakage inherent in contemporary split DNN execution techniques, as well as in various applications based on the extracted DNN features.

In this paper, we explore the use of diffusion models, a promising technique for image synthesis, to enhance feature inversion quality. We also investigate the potential of incorporating alternative forms of prior knowledge, such as textual prompts and cross-frame temporal correlations, to further improve the quality of inverted features. Our findings reveal that diffusion models can effectively leverage hidden information from the DNN features, resulting in superior reconstruction performance compared to previous methods. This research offers valuable insights into how diffusion models can enhance privacy and security within applications that are reliant on DNN features.

## 1  Introduction

---

[*]Work undertaken during time at Meta
[†]Work undertaken during time at Meta

Inverting visual features within DNNs presents a significant challenge in the realm of privacy for deep learning. The primary goal of feature inversion is to reverse the outputs (or intermediate results) of a pre-trained DNN and reconstruct the original image. This form of privacy attack, known as *feature inversion attack*, can raise privacy concerns across various domains. Modern systems that perform face recognition azu (2018); ama (2021); Schroff et al. (2015); Aggarwal et al. (2021); Bhat & Jain (2023); Lezama et al. (2017), AR/VR applications Ma et al. (2021); Fu et al. (2023); Zhang et al. (2021); Chu

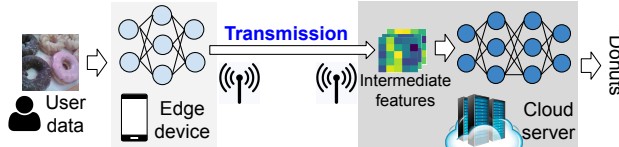

Figure 1: The DNN undergoes layer-wise partitioning and is divided between the edge and the cloud, with the intermediate features being transferred between them. In this example, we focus on the presence of a single edge device.

et al. (2020), and image or text retrieval Zhou et al. (2017); Lu et al. (2020; 2017); Song & Raghunathan (2020); Borgeaud et al. (2022) often store and process auxiliary data in the form of extracted features from the original input. For example, in a face recognition system, the human face is first encoded with a DNN encoder (e.g., FaceNet Schroff et al. (2015), CLIP Bhat & Jain (2023); Shen et al. (2023)) and the resultant feature vector is then searched over the database for identity matching via vector comparisons. Feature inversion attacks can be used to reconstruct the face of private users Mai et al. (2018).

Moreover, feature inversion attack also leads to a serious privacy leakage in the Split DNN computing paradigm Hauswald et al. (2014); Park et al. (2022); Kang et al. (2017); Teerapittayanon et al. (2016; 2017); Zhang et al. (2020a); Akintoye et al. (2023); Dong et al. (2022a); Karjee et al. (2022); Luo et al. (2023); Mubark et al. (2024); Lee et al. (2023); Feltin et al. (2023); Zeng et al. (2020); Ding et al. (2023); Kang et al. (2022); Matsubara et al. (2019; 2022); pys (2020); spl (2020). Within this paradigm, a layer-wise partitioning of the pretrained DNN into two or more blocks, aligning with the computational capabilities of the edge devices, as shown in Figure 1. During the execution, the user data is first processed using one or more local DNNs that contain the initial layers. The intermediate results are then transmitted to the central server for the execution of subsequent DNN layers. Split DNN computing has been widely adopted to accommodate the execution of increasingly large DNN on resource-constrained devices like mobile phones, and is believed to enhance user privacy by keeping user data on the local device—only the intermediate features are sent to the less secure cloud environment. However, this privacy enhancement turns out to be frail, as recent studies have shown that the intermediate features can be inverted via feature inversion attacks to reconstruct user inputs from the intermediate outputs of parts of the DNN Mahendran & Vedaldi (2015); Dosovitskiy & Brox (2016); He et al. (2019); Dong et al. (2021); Maeng et al. (2022); Song & Raghunathan (2020); Morris et al. (2023).

The broad applicability of feature inversion renders it a fundamental problem in ML security and privacy. On the other hand, feature inversion is not an easy task, particularly when dealing with features extracted from later layers of a network. Intuitively, the learned feature contains more high-level semantic information about objects in an image but less information about the raw input as depth increases. As a result, nearly all of the existing feature inversion methods fail when attempting to invert features from later layers of a deep network. This explains why much of the existing research concentrates on feature inversion for shallow DNNs with lower input resolutions Mahendran & Vedaldi (2015); Dosovitskiy & Brox (2016); He et al. (2019); Maeng et al. (2022).

The recent advancement of generative AI (GenAI) models opens up new possibilities to improve the quality of feature inversion attacks through their comprehensive understanding of image data distributions across real-world scenes. Among the multitude of existing GenAI techniques, Diffusion Models (DMs) Ho et al. (2020) have emerged as a remarkable breakthrough in generative modeling. Through extensive training with vast datasets comprising millions of real-world images, DMs obtain a high-quality, photorealistic image generation capability.

In this work, we demonstrate that recent advancements in DMs can be utilized to greatly enhance feature inversion. Instead of inverting DNN features directly to image pixels, we aim to recover the input vector in the latent space of a *latent diffusion model* (LDM) that, when converted to an image through reverse diffusion and forwarded through the DNN, matches the target DNN features. In addition, another noteworthy feature

of DMs is their capacity to take textual descriptions as input and produce synthetic outputs conditioned on these textual prompts. We also demonstrate that this capability enables the attackers [1] to specify the prior knowledge of a target image with natural language to utilize their existing knowledge about the victims, if available, by providing a textual description to DMs. Doing so enables the inversion of features that are much deeper into the network. Finally, in practice, as edge devices often process a continuous stream of input frames, we propose another variant that uses the temporal correlation in the features between consecutive input frames to enhance the reconstruction quality. Our main contributions are as follows:

- *Feature inversion using diffusion model prior.* We demonstrate that the exceptional image generation capabilities of DMs can be effectively employed to improve DNN feature inversion. We explore two threat models that closely describe the practical scenarios. To the best of our knowledge, this marks the first research endeavor showing the use of DMs for enhancing DNN feature inversion.

- *Incorporating textual prior for feature inversion.* We demonstrate that incorporating textual prior information about user inputs can significantly enhance the quality of feature inversion. To integrate this textual prior knowledge and achieve improved feature inversion quality, we introduce new training loss terms as a part of the inversion process.

- *Feature inversion for videos.* When processing a sequence of temporally correlated inputs, we show that feature inversion can be further enhanced by considering the temporal correlation among consecutive input frames.

- The evaluation results show that our approach exhibits significant superiority over the state-of-the-art approaches in feature inversion quality across a variety of evaluation metrics. For some backbone DNN models that are trained with self-supervised learning, we can achieve end-to-end inversion by reconstructing the input from the DNN outputs.

## 2 Background and Related Works

### 2.1 Diffusion Models

Diffusion models (DMs) Ho et al. (2020) have recently gained significant attentions for its remarkable ability to generate diverse photorealistic images. It is a parameterized Markov chain trained through Variational inference to generate samples that match the data distribution over a finite duration. Specifically, during the *forward process* of DMs, given an input image $x_0 \sim q(x)$, a series of Gaussian noise is generated and added to the $x_0$, resulting in a sequence of noisy samples $\{x_t\}, 0 \leqslant t \leqslant T$.

$$q(x_t|x_{t-1}) = \mathcal{N}(x_t; \sqrt{1 - \beta_t}x_{t-1}, \beta_t I) \tag{1}$$

where $\beta_t \in (0, 1)$ is the variance schedule that controls the strength of the Gaussian noise in each step.

During the *reverse process*, given a randomly sampled Gaussian noise $\mathcal{N}(x_T; 0, I)$, the synthetic images are generated progressively with the following procedure:

$$p_\theta(x_{t-1}|x_t) = \mathcal{N}(x_{t-1}; \mu_\theta(x_t, t), \hat{\beta}_t I) \tag{2}$$

where $\mu_\theta(x_t, t)$ and $\hat{\beta}_t$ are defined as follows:

$$\mu_\theta(x_t, t) = \frac{1}{\sqrt{\alpha_t}}(x_t - \frac{1 - \alpha_t}{\sqrt{1 - \bar{\alpha}_t}}\epsilon_{\theta,t}), \ \hat{\beta}_t = \frac{1 - \bar{\alpha}_{t-1}}{1 - \bar{\alpha}_t} \tag{3}$$

In Equation 3, $\epsilon_{\theta,t}$ denotes the predicted noise that is generated with a trained U-Net, $\alpha_t = 1 - \beta_t$, and $\bar{\alpha}_t = \prod_{i=1}^t \alpha_t$.

---

[1]In this paper, we will employ the terms "attacker" and "adversary" interchangeably.

Since their inception, there has been a variety of subsequent research that builds upon DMs. Several alternative approaches to accelerate the reverse process have been proposed Song et al. (2020a); Lyu et al. (2022); Lu et al. (2022). *Latent diffusion models* (LDMs) were introduced in Rombach et al. (2022) to perform the reverse process within the latent space of an autoencoder. The outcome of this reverse process is then fed to a decoder, which generates the synthetic images. LDMs offer a simple yet efficient way of enhancing both the training and sampling efficiency of LDMs without compromising their quality. LDMs can also be integrated with text encoders for text-to-image generation, as explored in Saharia et al. (2022); Rombach et al. (2022). Typically, these models include a pre-trained text encoder that takes user textual descriptions as input, effectively guiding the reverse process to generate the desired synthetic output.

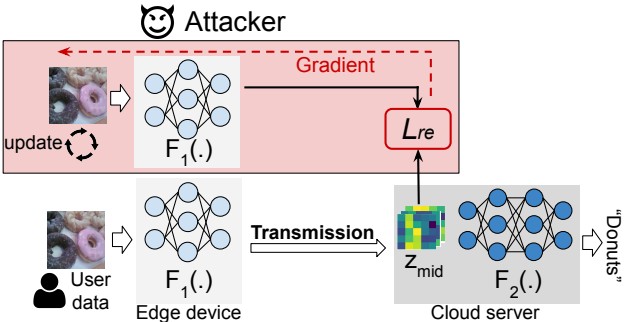

Figure 2: Feature inversion attack under white-box settings in a Split DNN execution scenario: The first-stage model $F_1(.)$ is exposed to the attacker. The attacker's goal is to generate an input image such that, when passed through $F_1(.)$, it produces intermediate features closely matching those of a target input.

## 2.2 Feature Inversion Attacks against DNNs

Feature inversion has been studied by various literature. Dosovitskiy & Brox (2016) showed that DNN features can be inverted by training a network to reconstruct the corresponding input images given their features. He et al. (2019) first demonstrated that feature inversion can lead to a leakage of private input for split DNN computation, and further showed that introducing the total variation loss Rudin et al. (1992) can greatly improve feature inversion quality. Dong et al. (2021) revealed that the exposure of batch normalization parameters can lead to a significant enhancement in feature inversion quality. Unsplit Erdoğan et al. (2022) operated within a black-box setting where the attackers lack knowledge of the model parameters, and developed techniques aimed at reconstructing both user inputs and model parameters. The study presented in Otroshi Shahreza & Marcel (2024) aims to create a facial image capable of deceiving facial recognition systems, without requiring the generated image to exactly match the user face.

Our research demonstrates the use of LDMs as a prior can significantly enhance feature inversion quality, an aspect not explored in prior studies. We also explore incorporating diverse prior knowledge sources, such as text and cross-frame correlations, to further improve reconstruction quality. These advanced techniques enable state-of-the-art feature inversion performance, surpassing prior methods. Then, we will discuss various application scenarios of feature inversion attacks.

## 2.3 Split DNN Computing

Split DNN computing has garnered significant attentions from both academia and industry, as evidenced by numerous studies Hauswald et al. (2014); Teerapittayanon et al. (2016); Kang et al. (2017); Teerapittayanon et al. (2017); Karjee et al. (2022); Luo et al. (2023); Mubark et al. (2024); Lee et al. (2023); Feltin et al. (2023); Zeng et al. (2020); Ding et al. (2023); Kang et al. (2022); Matsubara et al. (2019; 2022); Lin et al. (2024); Zhang et al. (2022); Dong et al. (2022b); Zhang et al. (2024). Additionally, solutions based on split learning and inference have been actively implemented and embraced across both commercial and open-source applications pys (2020); spl (2020). Among the multiple partition strategies Kang et al. (2017); Zhang et al. (2020a), layerwise partition has been widely employed Hauswald et al. (2014); Teerapittayanon et al. (2016); Kang et al. (2017); Teerapittayanon et al. (2017). This approach entails splitting the DNN into two or more parts and executing on multiple devices. The study by Hauswald *et al.* Hauswald et al. (2014), is among the initial research efforts that moved the later stages of image classification computation to cloud servers. Neurosurgeon Kang et al. (2017) and DDNN Teerapittayanon et al. (2017) introduced a technique for automatically distributing DNN models between a mobile device and a cloud server, considering factors like network latency and energy usage. Meanwhile, BranchyNet Teerapittayanon et al. (2016) made use of

early exit points within the DNN layers to enable adaptive DNN inference based on the input complexity, further reducing the processing latency.

## 2.4 Applications based on Extracted Features

Modern systems often store and process auxiliary data in the form of features extracted from the DNN encoder. For instance, in face recognition systems, the image of human face is initially encoded with a DNN, and the resulting feature vector is then searched over the database through vector comparisons azu (2018); ama (2021); Schroff et al. (2015); Aggarwal et al. (2021); Lezama et al. (2017).

In addition, some AR/VR tasks, such as Codec Avatar Ma et al. (2021); Richard et al. (2021); Fu et al. (2023), also rely on extracted features to operate. Codec Avatar is a high fidelity animatable human face model designed for the purpose of remotely sharing spaces with each other. To generate the Codec Avatar, an encoding process is first performed on the transmitter headset device: cameras linked to the VR headset capture partial facial images, which are then encoded by a DNN model into feature vectors and transmitted to the receiver headset device. On the receiver side, upon the reception of the feature vectors, the decoder reconstructs the avatar's geometry and appearance, enabling the real-time rendering of the transmitter's photorealistic face.

Finally, in the field of image and text retrieval, recent studies have advocated for the adoption of vector database services to facilitate scalable embedding matching and retrieval, yielding enhanced performance Zhou et al. (2017); Lu et al. (2020; 2017); Song & Raghunathan (2020); Borgeaud et al. (2022). To operate, the data owner transmits only embeddings of the raw data from the DNN encoder, to the third-party service, without revealing the actual text content. Subsequently, the database server returns a search result, indicating the index of the matching document on the client side.

## 3 Threat Models

We begin by first describing the underlying threat model for our feature inversion attack. We consider two settings, white-box and black-box, for the two variants of our attack.

### 3.1 Threat Model for White-Box Settings

We focus on a scenario where the target model $F(.)$ is divided into two parts: $F(.) = F_2 \circ F_1$. Here, we use $x_{gt}$ to represent the user input and $z_{mid}$ to denote the intermediate feature. We assume that $F_1$, referred to as the *user model*, is executed within a secure environment such as an edge device, where internal computations are protected from external access. In contrast, $F_2$ runs in an untrusted environment (e.g., a public cloud), where the input $z_{mid} = F_1(x_{gt})$ may be exposed to potential adversaries. This split computing setup is realistic and widely adopted in practice, particularly for latency-sensitive and privacy-aware applications. In such systems, it is common for early layers (i.e., shallow layers) to be executed on-device to reduce bandwidth and protect raw input data, while the remaining layers are offloaded to the cloud. Therefore, accessing shallow-layer outputs like $z_{mid}$ is both feasible and practical under this model, especially when the communication between edge and cloud is not end-to-end encrypted or when decryption is required before cloud-side processing.

Although encryption can be applied during transmission, we assume that $z_{mid}$ must be decrypted before execution in $F_2$, which reintroduces exposure risk. In this setting, adversaries may attempt to reconstruct the original input. In the white-box scenario, we assume the attacker has full access to the structure and parameters of $F_1$, but not to the input $x_{gt}$ or intermediate activations within $F_1$. Our goal is to reconstruct an input that produces intermediate outcomes similar to the observed $z_{mid}$, which can be formulated as follows:

$$x_{re} = \underset{x}{\arg\min} \; \mathcal{L}_{re}(F_1(x), z_{mid}), \tag{4}$$

where $\mathcal{L}_{re}(.,.)$, referred to as the *reconstruction loss*, represents the loss function employed for measuring similarity, with the $l_2$ distance being used in this study. We construct $x$ to minimize the loss function as

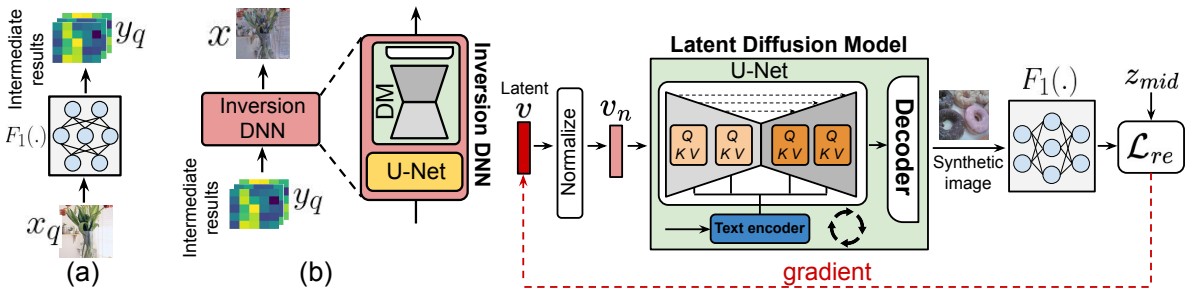

Figure 3: Black-box feature inversion attack procedure: The diffusion model takes the intermediate features as input and generates a reconstructed image.

Figure 4: Feature inversion using a latent diffusion model, where the model takes an input vector $v$ and generates synthetic images that closely resemble the victim image.

illustrated in Equation 4 (Figure 2). Previous studies He et al. (2019); Maeng et al. (2022) have demonstrated the feasibility of achieving high-fidelity input reconstruction when $F_1$ is shallow and the input dimension is limited. However, as $F_1$ gets deeper, an increasing portion of information within $x_{gt}$ is filtered out by DNN operations, such as pooling layers, retaining only the essential task-related information. This greatly complicates the task of feature inversion.

## 3.2 Threat Model for Black-Box Settings

In the black-box variant of the feature inversion attack, the threat model resembles that of the white-box approach, with the key distinction being the relaxation of the assumption regarding the adversary's knowledge of $F_1(.)$. Here, the adversary can only gather information about $F_1$ indirectly through querying it. As a result, the adversary gains access to the input queries and their corresponding outputs from $F_1(.)$, as shown in Figure 3(a). Denote $X_q = \{x_q\}$ a set of input queries sent by the adversary and $Y_q = \{y_q\}$ the corresponding outputs from $F_1(.)$. Next, an inversion DNN $F_\theta^{inv}(.)$ is trained to take $Y_q$ as input and generate $X$ that closely resembles $X_q$ (Figure 3 (b)), namely:

$$\min_\theta \sum_{(x_q,y_q)} \mathcal{L}_{re}(F_\theta^{inv}(F_1(x_q)), x_q) \tag{5}$$

For a black-box attack, the attacker only needs access to the function $F_1(.)$ by querying it with their own inputs and collecting the outputs for training. The internal architecture of $F_1(.)$ does not need to be known.

## 3.3 Generalizability of the Threat Model

Our threat model described in Section 3.1 and Section 3.2 can also be applied to systems involving more than two participants. However, given that many real systems are typically divided into two parties Teerapittayanon et al. (2016; 2017); Kang et al. (2017), we focus on a two-participant system for the remainder of this paper, without sacrificing generality. Additionally, by making $F_1 = F$ and $F_2 =$, our approach can also be applied to the scenario of end-to-end feature inversion, where the objective is to invert the DNN output to reconstruct the input. This presents a significant privacy concern for applications that operate based on extracted features, such as face recognition, Codec Avatar, etc.

## 4 White-box Feature Inversion

In this section, we describe the white-box attack methodologies in details. Particularly, we first present our methodology for feature inversion with LDMs. Subsequently, we investigate the impact of textual priors on feature inversion in Section 4.1 and discuss multi-frame reconstruction in Section 4.2.

We leverage the prior knowledge embedded within the LDM to reconstruct the user input $x_{gt}$. Let $D(v, e)$ represent the generating function of the LDM. Here, $v$ denotes the input latent variable, which is expected

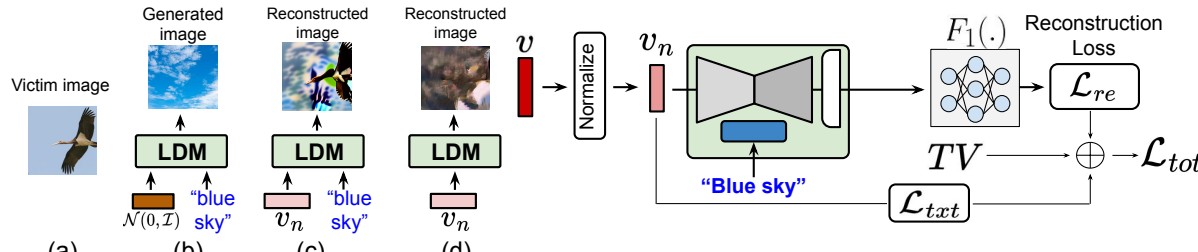

(a) (b) (c) (d)

Figure 5: Impact of textual prior.

Figure 6: White-box feature inversion with textual prior on the diffusion model.

---

**Algorithm 1** Feature Inversion with LDMs

---

$F_1(.)$ is the user DNN model.
$v$ is the input latent vector of LDMs.
$I$ is total number of iterations.
$z_{mid}$ is the intermediate result.
$\epsilon$ is the learning rate.
**for** $1 \leqslant i \leqslant I$ **do**
$\quad v_n^i = \frac{v^i - \text{mean}(v^I)}{\text{std}(v^i)}$
$\quad \mathcal{L}_{tot} = ||F_1(D(v_n^i)) - z_{mid}||^2 + \lambda_s TV(D(v_n^i))$
$\quad v^{i+1} = v^i - \epsilon \frac{d\mathcal{L}_{tot}}{dv}$
$\quad i = i + 1$
$v_n = \frac{v^I - \text{mean}(v^I)}{\text{std}(v^I)}$
**return** $D(v_n)$.

---

to follow a normal distribution Ho et al. (2020), and $e = E(t)$ represents the text embedding. The function $E(.)$ indicates a pre-trained text encoder, and $t$ corresponds to the text input provided by the user. In this section, we ignore the text prompt by setting the text embedding $e$ to a vector of zeroes, and will examine the influence of the text prior in Section 4.1. We then search for the input latent variable $v$ that allows the LDM to produce a synthetic output, denoted as $D(v_n)$. This output, when passed to $F_1(.)$, will result in a similar intermediate output as $z_{mid}$ (Figure 4).

$$v^* = \arg\min_v \ \mathcal{L}_{re}(F_1(D(v_n)), z_{mid}) + \lambda_s TV(D(v_n)) \tag{6}$$

As the LDM necessitates input data to approximate a normal distribution for photorealistic image generation, we implement a soft restriction on the variable $v$ by normalizing it prior to forwarding it to the LDM. Specifically, we define $v_n = \frac{v - \text{mean}(v)}{\text{std}(v)}$ as the normalized version of $v$, which serves as the input for the LDM. We observe that applying normalizing operation can greatly enhance the feature inversion performance. $z_{mid} = F_1(x_{gt})$ is the intermediate result generated from the user input. $TV(.)$ represents the *Total Variation* Rudin et al. (1992) which is used to reduce the abrupt pixel variations across the reconstructed image. $TV(x)$ is defined as follows:

$$TV(x) = \frac{1}{MN} \sum_i \sum_j (|x_{i+1,j} - x_{i,j}|^2 + |x_{i,j+1} - x_{i,j}|^2) \tag{7}$$

where $M$ and $N$ represent the spatial size of the image, and $\lambda_s$ denote the weight of the TV loss. The feature inversion process is summarized in Algorithm 1.

## 4.1 White-box Inversion with Textual Prior

Another important characteristic of LDMs is their ability to take text prompt as input and produce synthetic outputs guided by textual descriptions. We leverage this capability in our feature inversion attacks by allowing the attacker to express their prior knowledge about the user input in the form of natural language.

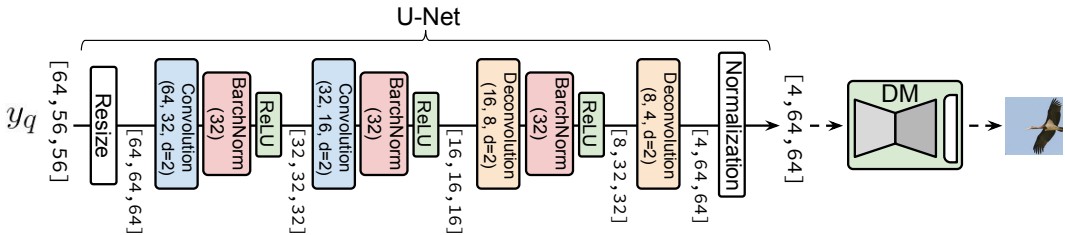

Figure 7: A sample architecture of U-Net. The dimension will adjust based on the dimension of $y_q$. In this example, the intermediate result $y_q$ is of dimension $64 \times 56 \times 56$, with a batch size of 1. The LDM input has a dimension of $4 \times 64 \times 64$. $d$ in the (de)convolutional blocks denotes the stride.

Different from generic image priors such as total variation, this form of text prior can be specific to each target image and further enhances the quality of feature inversion using diffusion models.

To incorporate the text prior into the feature inversion process, consider the private user image depicted in Figure 5 (a). Assuming the adversary possesses prior knowledge of this private image, they will convey this knowledge to the LDM through textual description. The LDM will take these textual description together with another randomly generated, normally-distributed input to produce an image visually akin to the user image (Figure 5 (b)). Subsequently, we proceed to further train the LDM input $v_n$ to enhance the LDM's ability to refine the output, making it more closely resemble the user image $x_{gt}$ (Figure 5 (c)). The resulting reconstructed image has a much better quality than that reconstructed without a textual description, which is shown in Figure 5 (d).

Considering that the normally-distributed latent coupled with the text prior can produce an output that relates to user input, we utilize this insight by further pushing $v_n$, the input of LDM, to approach a random variable generated from a normal distribution. To achieve this, we assess the Gaussianity of $v_n$ using the negentropy metric outlined in Hyvärinen & Oja (2000), resulting in an additional loss term denoted as $\mathcal{L}_{txt}$, defined as follows:

$$\mathcal{L}_{txt} = -\mathbb{E}\left[\frac{1}{\alpha^2}logcosh^2(\alpha v_{n,i})\right] \tag{8}$$

where $1 \leqslant \alpha \leqslant 2$ is a hyperparameter, the expectation $\mathbb{E}(.)$ is taken over the elements of $v_n$. The total loss $\mathcal{L}_{tot}$ can be described as:

$$\mathcal{L}_{tot} = \mathcal{L}_{re}(F_1(D(v_n, e)), z_{mid}) + \lambda_s TV + \lambda_{txt}\mathcal{L}_{txt}(v_n) \tag{9}$$

where $e = E(t)$ represents the embedding of the textual description, which serves as an additional input to the LDM, $\lambda_{txt}$ is the weight factor to balance the loss terms (Figure 6). The remaining inversion algorithm is similar to Algorithm 1. The detailed algorithm is given in the appendix.

## 4.2 White-box Multi-frame Reconstruction

In this section, we explore the problem on multi-frame feature inversion. This scenario closely resembles real-world situations where edge devices handle a continuous stream of input frames, such as burst mode photos or video clips. In this context, the local DNN processes consecutive input frames that exhibit temporal correlation, the intermediate features are then transmitted to cloud servers for subsequent processing. The goal is to reconstruct the entire input image sequence using the intermediate results.

In particular, consider $x_{gt,k}$, where $1 \leqslant k \leqslant K$, to represent a sequence of $K$ user inputs. Additionally, let $z_{mid,k}$ and $v_k$ represent the corresponding local DNN output and input latent variable for $x_{gt,k}$. We introduce an additional loss component $\mathcal{L}_c(.)$ aimed at minimizing the disparity among the latent vectors $v_k$ across these frames. To achieve this, the multi-frame reconstruction process can be realized by solving the following optimization problem:

$$\min_{v_k, 1\leqslant k \leqslant K} \sum_{k=1}^{K}\left[\mathcal{L}_{re,k} + \lambda_s TV_k + \lambda_c\mathcal{L}_c(v_k, \bar{v})\right], \tag{10}$$

---

**Algorithm 2** Multi-frame Feature Inversion

---

$F_1(.)$ is the user DNN model.
$K$ is the total number of input frames.
$v_k^m$ is the latent vector of input k at iteration m.
$z_{mid,k}$ is the intermediate results for input k.
$M$ is total number of iterations.
$\epsilon$ is the learning rate.
$\lambda_s, \lambda_c$ are the weights for TV loss and temporal loss, respectively.

**for** $1 \leqslant m \leqslant M$ **do**
$\quad \bar{v}^m = \frac{1}{K} \sum_k v_k^m$
$\quad \mathcal{L}_{tot} = 0$
$\quad$ **for** $1 \leqslant k \leqslant K$ **do**
$\quad\quad v_{k,n}^m = \frac{v_k^m - \text{mean}(v_k^m)}{\text{std}(v_k^m)}$
$\quad\quad \mathcal{L}_{tot} + = ||F_1(D(v_{k,n}^m)) - z_{mid,k}||^2 + \lambda_s TV(D(v_{k,n}^m)) + \lambda_c ||v_k^m - \bar{v}^m||^2$

$\quad$ Compute the gradients based on $\mathcal{L}_{tot}$.
$\quad$ **for** $1 \leqslant k \leqslant K$ **do**
$\quad\quad v_k^{m+1} = v_k^m - \epsilon \frac{d\mathcal{L}_{tot}}{dv_k}$
$\quad\quad m = m + 1$

**for** $1 \leqslant k \leqslant K$ **do**
$\quad v_{k,n}^M = \frac{v_k^M - \text{mean}(v_k^M)}{\text{std}(v_k^M)}$
$\quad$ **return** $D(v_{k,n}^M)$.

---

where $\bar{v} = \frac{1}{K}(\sum_{k=1}^K v_k)$ represents the average of the input latent vectors across the $K$ frames. The loss function $\mathcal{L}_c$ is utilized to minimize the disparity between the latent vectors for each frame. In this study, we have observed that simply minimizing their $l_2$ distance yields an excellent reconstruction quality. The parameter $\lambda_c$ serves as a weight to balance the importance of these two loss functions. $\mathcal{L}_{re,k}$ and $TV_k$ denote the reconstruction loss and total variation loss for reconstructing $x_{gt,k}$. The detailed algorithm is given in Algorithm 2.

## 5 Black-box Feature Inversion

For feature inversion attacks with black-box settings, the attacker obtains access to the input queries and their corresponding outputs from $F_1(.)$, as depicted in Figure 3 (a). Let $X_q = \{x_q\}$ denote a set of input queries sent by the adversary and $Y_q = \{y_q\}$ represent the corresponding outputs from $F_1(.)$. The adversary then proceeds to train an inversion DNN $F_\theta^{inv}(.)$ designed to take $y_q$ as input and generate $x$ that closely resembles $x_q$ (Figure 3 (b)).

$F_\theta^{inv}(.)$ consists of two major components: a pre-trained LDM and an U-Net, which are denoted as $D(.)$ and $F_u(.)$, respectively. During the execution, $F_u(.)$ takes the intermediate data $y_q$ and generates the input latent variable for the LDM, which then produces the result $x = D(F_u(y_q))$. The training of the inversion DNN model involves minimizing the following loss function:

$$\theta_u^* = \arg\min_{\theta_u} \sum_{(x_q, y_q) \in \{(X_q, Y_q)\}} \mathcal{L}_{re}(D(F_u(y_q)), x_q) + \lambda_s TV \tag{11}$$

where $\theta_u$ represents the parameters of $F_\theta^{inv}(.)$. TV loss is introduced over the reconstructed input $D(F_u(y_q))$.

The architecture of U-Net is illustrated in Figure 7. During the forward pass, the input $y_q$ is first resized spatially. Following this, the intermediate output traverses through several blocks consisting of (de)convolutional layers, batch normalization layers, and ReLU layers. The resulting output is normalized before being sent to the diffusion model for image generation. The specific dimensions of the inversion DNN will vary depending on the shape of $y_q$.

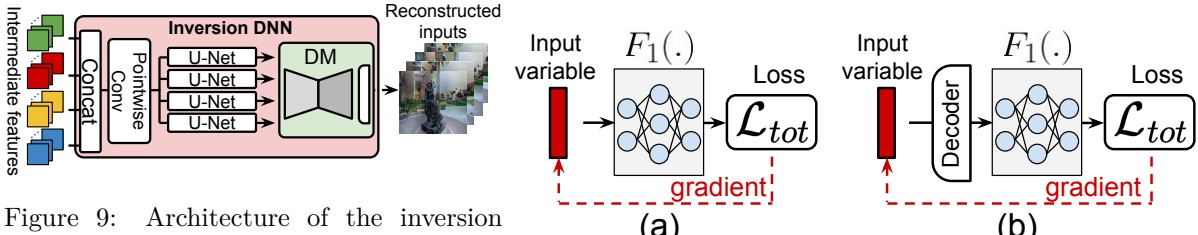

Figure 9: Architecture of the inversion DNN for black-box multi-frame reconstruction. In this example, the inversion DNN can generate four frames.

Figure 10: Baseline feature inversion algorithms.

### 5.1 Black-box Inversion with Textual Prior

Similar to the incorporation of textual priors to enhance reconstruction quality in the white-box setting, integrating textual priors into the inversion DNN can also improve the quality of the black-box feature inversion. The training procedure is outlined in Figure 8. The U-Net output is directed into the DM for image generation, while an additional loss function $\mathcal{L}_{txt}(.)$ is simultaneously applied to enhance its gaussianity. The overall loss function is shown as follows:

$$\theta_u^* = \arg\min_{\theta_u} \sum_{(x_q, y_q)} \mathcal{L}_{re}(D(F_u(y_q), e_q), x_q) + \lambda_s TV + \lambda_{txt}\mathcal{L}_{txt}(F_u(y_q))$$

(12)

Figure 8: Feature inversion with textual prior under black-box settings.

where $\mathcal{L}_{txt}(.)$ is the loss term that enforces the LDM inputs, $F_u(y_q)$, follows a gaussian distribution. $e_q = E(t_q)$ is the embeddings of the textual description $t_q$ that describes $x_q$.

### 5.2 Black-box Multi-frame Reconstruction

In this section, we explore the problem on multi-frame feature inversion under black-box settings. In particular, assume a group of consecutive frames with a total of $K$ images, the inversion DNN will take the intermediate results $Y_q^g = \{y_{q,k\in K}\}$ from each frame $k$ within this group $g$, and produce the $K$ outputs that will serve as the inputs of the LDM.

In order to exploit the temporal correlation within the intermediate results $Y_q^g$, we introduce a pointwise convolutional layer into the inversion DNN, as depicted in Figure 9. This pointwise convolutional layer will incorporate a weight filter with a spatial size of $1 \times 1$, enabling the learning of temporal correlations among the intermediate results $y_{q,k}$. The output of the pointwise convolution will then be separated into $K$ pieces each of which corresponds to one input frame, the outputs will be forwarded to the U-Net, whose architecture is shown in Figure 7. Each of the four U-Net will share its weights. The outputs from U-Net will further be delivered to the LDM, which will then reconstruct $x_{q,k}$ for each $k \in K$. The loss function for training is shown as follows:

$$\min_{\theta_u} \sum_g \left[ \mathcal{L}_{re}(D(F_u(Y_q^g)), X_q^g) + \sum_{k\in K} \lambda_s TV_k^g \right]$$

(13)

where $X_q^g = \{x_{q,k\in K}\}$ are the groups of ground-truth consecutive frames, and $TV_k$ is the TV loss of the $k$-th reconstructed frame within group $g$.

## 6 Evaluation Results for White-box Inversion

In this section, we present detailed evaluation of the white-box feature inversion technique described in Section 4. We first evaluate the quality of the inverted features over different applications in Section 6.2,

Section 6.4 and Section 6.5. Next, we explore the influence of the textual context in Section 6.6 and the multi-frame reconstruction in Section 6.7. Lastly, we conduct an ablation studies in Section 6.8 and Section 6.9.

## 6.1 Experiment settings

**Datasets and models:** We assess our feature inversion approach outlined in Algorithm 1 on ImageNet Deng et al. (2009) and COCO Lin et al. (2014) datasets. We employ various DNN architectures pre-trained on ImageNet as the target models for feature inversion, including ResNet-18, ResNet-50, and Vision Transformer (ViT). All of the pretrained models are downloaded from the official Pytorch website. Due to institutional restriction, we are unable to use the public latent diffusion model for publishing our research outcomes. Instead, we employed an LDM with an architecture highly similar to Stable Diffusion 2.1 Rombach et al. (2022) in terms of architecture, model size and pretraining techniques. Our internal model was pretrained on the dataset collected by a third party (Shutterstock) that is not public. Regarding the dataset, it consists of 385 million images: 321 million without people and 64 million with people. None of the victim images used in the feature inversion attack are included in the training dataset of the diffusion model. In addition, we have also previously conducted extensive experiments with the public Stable Diffusion 2.1, which yielded similar (and even better) results in terms of IS, PSNR, and SSIM scores than the results in this paper.

**Hyperparameters:** We set all $\lambda_s$ to 1 for the reconstruction loss defined in Equation 6, Equation 9, and Equation 13. The reconstruction process continues for a total of $T = 1500$ iterations. We adopt the Adam optimizer with an initial learning rate of 0.1, $\beta=(0.9,0.999)$. To expedite the reverse procedure, we configure the sampling steps of LDM to be 20 with a linear schedule Ho et al. (2020). We find that using 20 sampling steps can already yield high-quality feature inversion results. We investigate the impact of the sampling steps in Section 6.8. More evaluation results can be found in the appendix.

**Baselines:** We consider two algorithms for comparison. The first approach, referred to as *Direct Optimization (DO)*, reconstructs the input by directly optimizing Equation 4 over the image pixel space (Figure 10 (a)). This method has been utilized for input reconstruction in prior works He et al. (2019); Maeng et al. (2022); Erdoğan et al. (2022) and serves as the baseline to assess the impact of LDMs on the feature inversion. The second approach, known as the *Decoder-based (DB)* approach (Figure 10 (b)), employs only the LDM decoder for input reconstruction. Evaluating this approach helps us understand the influence of the iterative reverse process in LDMs on feature inversion attacks. It is worth noting that this **Decoder-based approach has not been investigated in prior works on feature inversion either**, but similar techniques using GAN decoders have been used in the context of gradient inversion Jeon et al. (2021); Li et al. (2022) and model inversion Zhang et al. (2020b). Finally, we denote our method as the *DM-based (DMB)* approach.

## 6.2 Feature Inversion on Split Models for Image Classification

We first assess the reconstruction quality of our feature inversion attacks *without* text prior. We randomly select 100 images from the ImageNet and COCO test datasets and feed them to a pre-trained ResNet-18, ResNet-50 and ViT-base model, respectively. For each of these target DNN models, we evenly divide them into blocks of layers and extract intermediate results at the end of each block. Subsequently, we employ the techniques outlined in Section 4 to reconstruct the user input.

**Qualitative result:** Figure 11 depicts the feature inversion results for ResNet-50 over ImageNet, respectively. The original image is displayed in the left column for reference. DM-based method consistently demonstrates superior reconstruction qualities across all datasets and DNN architectures. Notably, our approach achieves high-quality input reconstructions, even when utilizing features from very deep layers (e.g., layer 36 in ResNet-50), whereas other baseline methods struggle to achieve comparable performance.

**Quantitative result:** To quantify the quality of the reconstructed images, we utilize three metrics. The first metric is Inception Score (IS) Salimans et al. (2016), which is commonly used to evaluate the quality of image generation in prior works Song et al. (2020b); Xu et al. (2019); He et al. (2019); Dong et al. (2021). For instance, generative AI models like StackGAN Zhang et al. (2017) and GAN-INT-CLS Reed et al. (2016)

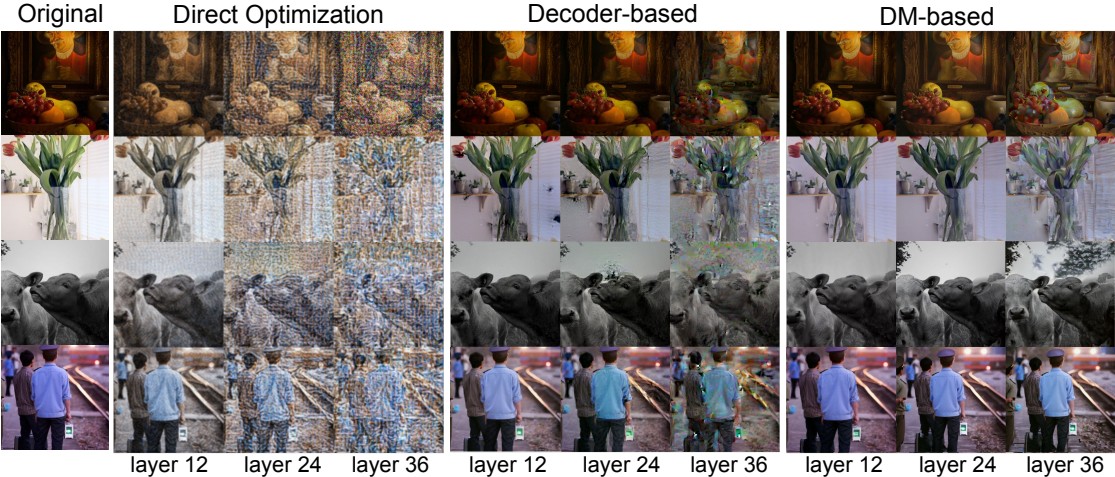

Figure 11: Feature inversion over ResNet-50 on ImageNet, features are extracted from the end of layer 12, layer 24 and layer 36 respectively.

typically generate images with IS scores around 3 to 5, while diffusion models can achieve IS scores as high as 10 Ho et al. (2020).

The second metric is Peak Signal-to-Noise Ratio (PSNR), which calculates the ratio between the maximum possible value of a signal and the power of distorting noise affecting the quality of its representation. The mathematical expression for PSNR is as follows:

$$PSNR(I^{ori}, I^{re}) = 10log(\frac{255}{\frac{1}{MN}\sum_{i=1}^{M}\sum_{j=1}^{N} I_{i,j}^{ori} - I_{i,j}^{re}}) \tag{14}$$

where $I^{ori}$ and $I^{re}$ denote the original and reconstructed images, and they both have a size of $M \times N$. 255 is the maximum pixel value. PSNR is a commonly used metric for assessing image quality, particularly when comparing a compressed or reconstructed image to its original version. It is frequently employed in the image processing tasks to quantify the degree of distortion introduced. PSNR values between 30-50 dB are typically considered indicative of excellent image quality.

Lastly, the Structural Similarity Index Measure (SSIM) Wang et al. (2004) is a metric used to assess image quality by comparing the similarity between two images. In our scenario, we evaluate SSIM between the reconstructed images and the original image. Unlike the PSNR, which focuses solely on pixel differences, SSIM evaluates changes in structural information, luminance, and contrast, making it more closely aligned with human visual perception. SSIM values range from 0 to 1, with higher values indicating better image quality. Table 1 gives the mean values of average IS, PSNR and SSIM across 100 reconstructed images over ImageNet for the various model architectures and feature inversion methods. We can observe three trends:

- For the same model-layer pair (i.e. each column), DM-based method (DMB) achieves the highest average IS/PSNR/SSIM compared to the Direct Optimization (DO) and Decoder-based (DB) approaches. This shows that our diffusion-based feature inversion attack is highly effective.

- For the same model, inverting features extracted from later layers results in lower IS/PSNR/SSIM for all three methods. However, with our diffusion-based method, the reduction in reconstruction quality is much less pronounced, showing that the attack is more capable of inverting later layer features.

- In comparison, inverting features for ViT models is more challenging, although it remains feasible to invert features using output from middle layers of ViT (e.g., layer 5).

Table 1: Evaluation results: 'DO', 'DB', and 'DMB' refer to direct optimization, decoder-based, and DM-based approaches. 'L' is the feature extraction layer. PSNR is shown in db. For IS, PSNR and SSIM, higher values indicate better results.

| Metric | Method | ResNet-18 | | | ResNet-50 | | | ViT-base | | |
| --- | --- | --- | --- | --- | --- | --- | --- | --- | --- | --- |
| | | L4 | L8 | L12 | L12 | L24 | L36 | L3 | L4 | L5 |
| IS | DO | 5.63 | 3.92 | 1.40 | 5.55 | 3.88 | 1.28 | 5.46 | 3.95 | 1.63 |
| | DB | 6.84 | 5.97 | 4.20 | 6.90 | 5.86 | 4.38 | 6.76 | 5.76 | 3.93 |
| | DMB | 7.23 | 6.86 | 6.48 | 7.36 | 6.90 | 6.55 | 7.14 | 6.77 | 6.58 |
| PSNR | DO | 29.3 | 14.6 | 9.51 | 28.9 | 15.3 | 8.04 | 28.5 | 17.6 | 9.42 |
| | DB | 35.2 | 32.6 | 18.6 | 35.4 | 33.0 | 18.3 | 36.8 | 33.1 | 19.9 |
| | DMB | 41.0 | 36.3 | 29.1 | 40.2 | 37.0 | 29.9 | 42.6 | 38.9 | 32.5 |
| SSIM | DO | 0.87 | 0.58 | 0.13 | 0.85 | 0.62 | 0.09 | 0.87 | 0.66 | 0.11 |
| | DB | 0.93 | 0.90 | 0.72 | 0.92 | 0.88 | 0.70 | 0.93 | 0.90 | 0.75 |
| | DMB | 0.97 | 0.94 | 0.86 | 0.96 | 0.95 | 0.88 | 0.98 | 0.94 | 0.91 |

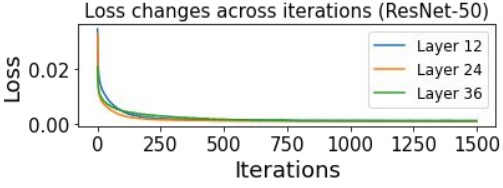

Figure 12: Changes on training loss.

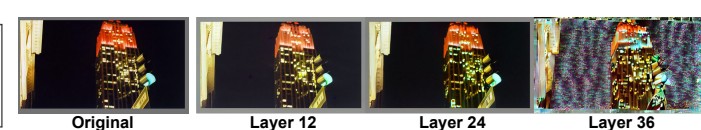

Figure 13: White-box feature inversion from different layers of YOLO.

In Figure 23, we illustrate the changes on the loss function throughout the reconstruction process as outlined in Algorithm 1 using the intermediate features from ResNet-50. It is evident that the loss values converge by the end of the 1500 iterations.

## 6.3 Comparison with White-box GAN-based Inversion Method

We perform another experiment by replacing the diffusion model with StyleGAN3 Karras et al. (2021). To achieve this, we replace the diffusion model in Figure 4 of the paper with the StyleGAN3, and train the latent vector of StyleGAN3 to reconstruct the user input. We evaluate under both white-box setting. All training settings remain consistent with those described in Sections 6.1 of the paper. The results are shown in Table 2 for white-box and black-box settings, respectively. We can see that our diffusion model-based approach outperforms the GAN-based approach.

Table 2: Evaluation results of white-box setting. "StyleGAN-based" and "DMB" denote StyleGAN-based and DM-based approaches, respectively. "L" denotes the layer where the features are extracted. PSNR is shown in db. For IS, PSNR and SSIM, higher values indicate better results.

| Metric | Method | ResNet-18 | | | ResNet-50 | | | ViT-base | | |
| --- | --- | --- | --- | --- | --- | --- | --- | --- | --- | --- |
| | | L4 | L8 | L12 | L12 | L24 | L36 | L3 | L4 | L5 |
| IS | StyleGAN-based | 7.02 | 6.22 | 5.06 | 7.04 | 6.12 | 4.88 | 6.89 | 5.92 | 4.00 |
| | DMB | 7.23 | 6.86 | 6.48 | 7.36 | 6.90 | 6.55 | 7.14 | 6.77 | 6.58 |
| PSNR | StyleGAN-based | 37.3 | 34.1 | 22.5 | 36.9 | 34.3 | 21.7 | 35.3 | 34.7 | 21.0 |
| | DMB | 41.0 | 36.3 | 29.1 | 40.2 | 37.0 | 29.9 | 42.6 | 38.9 | 32.5 |
| SSIM | StyleGAN-based | 0.94 | 0.92 | 0.78 | 0.94 | 0.90 | 0.76 | 0.92 | 0.88 | 0.74 |
| | DMB | 0.97 | 0.94 | 0.86 | 0.96 | 0.95 | 0.88 | 0.98 | 0.94 | 0.91 |

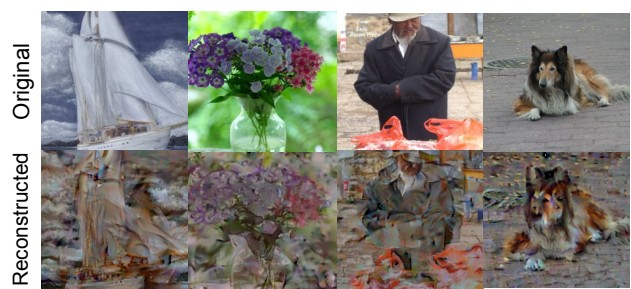

Figure 15: End-to-end feature inversion over CLIP image encoder.

## 6.4 Feature Inversion Results over YOLO

We present additional feature inversion results over a YOLO-v2 model based on ResNet-50 for object detection (Figure 13). We select 100 samples from the COCO dataset for the inversion experiment. Specifically, for YOLO, our method can achieve higher average inception scores of 8.13, 7.22, and 6.32 using features from layers 12, 24, and 36. This is much higher than IS scores obtained by DO method (5.60, 3.75 and 1.29 for L12, L24 and L36, respectively) and DB method (6.96, 5.80 and 4.47 for L12, L24 and L36, respectively).

## 6.5 End-to-end Feature Inversion over CLIP

In this section, we present the results for the **end-to-end feature inversion** over the CLIP Radford et al. (2021) image encoder. CLIP is language-visual multimodal DNN capable of understanding images and text jointly in a zero-shot manner, without the need for fine-tuning on a specific task. It aligns natural language prompts with images to perform a wide range of tasks, including image classification, image generation, and image-text retrieval. Specifically, the CLIP image encoder has been widely adopted for various of computer vision tasks including face

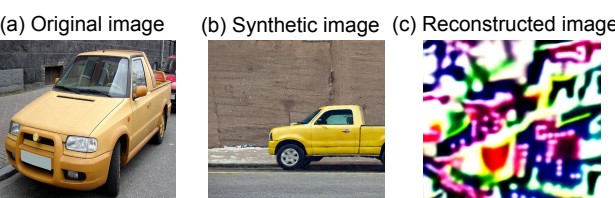

Figure 14: Limitations on utilizing text priors.

recognition Bhat & Jain (2023); Shen et al. (2023), image segmentation Wang et al. (2022b) and emotion classification Bondielli et al. (2021); Deng et al. (2022). We use the methodology described in Section 3.1 to invert the output features from the pretrained CLIP image encoder. Specifically, we download *clip-vit-base-patch32* from the huggingface official website hug (2021), and apply the settings described in the beginning of Section 6 for evaluation. We select 100 images from the ImageNet test dataset, some images together with their reconstructed versions are shown in Figure 15. Specifically, we obtain an IS, PSNR and SSIM of 3.54, 13.2 and 0.50, respectively. In comparison, DO method achieves a IS, PSNR and SSIM of 0.88, 5.78 and 0.11, respectively. Similarly, the DB method also attains IS, PSNR, and SSIM scores of 2.21, 9.74 and 0.33, respectively. Furthermore, we note that, when compared with a DNN designed for image classification, CLIP is much easier to invert at the same layer depth. This enhancement could be attributed to CLIP's tendency to retain more original image data to support a wide range of downstream tasks, thereby aiding feature inversion. In contrast, DNNs for image classification typically discard redundant information, preserving only the essential data required for recognizing object classes within the image.

## 6.6 Impact of Text Prior on Feature Inversion

In this section, we evaluate the impact of the text prior on the feature inversion results. Particularly, we extract intermediate features from deeper layers, such as layer 48 in ResNet-50. When attempting to invert these features from the deep layers, we observe a substantial quality degradation over the reconstructed input with the methods in Section 3.1. This decline can be primarily attributed to the loss of critical information embedded within deep features.

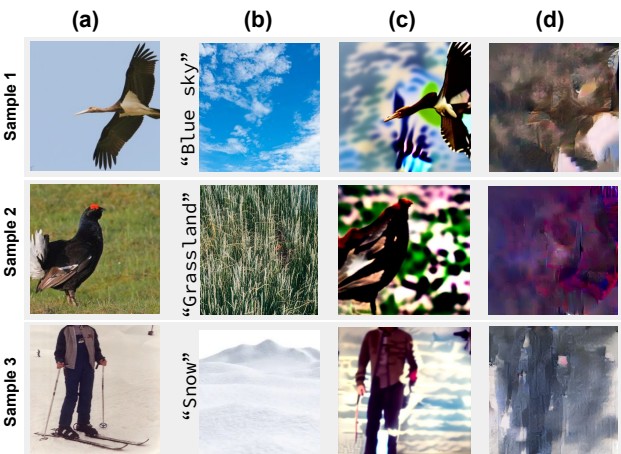

Figure 16: Impact of the text prior on feature inversion. Three samples with different scenes are selected from the ImageNet, shown in column (a). Column (b) shows the textual inputs and the corresponding synthetic images. Column (c) and (d) depict the reconstructed input with and without the text description.

To evaluate the impact of textual prior, we select multiple images from the ImageNet dataset featuring simple backgrounds, including scenes with sky, grassland, and snow (Figure 16 (a)). For each of these three background categories, we choose 10 images. We utilize the textual descriptions "blue sky", "a piece of grassland", and "snow" as textual guidance for the LDMs during the feature inversion process. Subsequently, these selected images are forwarded through ResNet-18, ResNet-50 and ViT-based, and we extract the results from the output of layer 16 for ResNet-18, layer 48 for ResNet-50, and layer 8 for ViT-base, respectively. The extracted intermediate results are then employed to reconstruct the input images using the techniques outlined in Section 4.1. Throughout the reconstruction process, $\lambda_s$ and $\lambda_{txt}$ in Equation 9 are set to 1 and 10, respectively. The rest of the settings remain the same as those in Section 6.2.

The reconstruction results are shown in Figure 16. We notice a significant enhancement in the feature inversion quality when incorporating the textual prior, as depicted in Figure 16 (c). These reconstructions, although not pixel-perfect, closely resemble the original images in (a) in terms of the semantic content. In comparison, the reconstructions obtained without textual prior, as shown in Figure 16 (d), are not semantically meaningful. Ta-

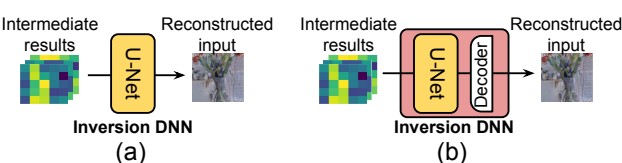

Figure 17: Baselines for black-box evaluation.

ble 3 shows quantitative results for feature inversion with (right) and without (left) textual prior in terms of average IS, PSNR and SSIM. Evidently, including the textual prior significantly improves the reconstruction quality quantitatively as well.

While the use of a textual prior can greatly enhance the reconstruction quality, it should be employed judiciously, as its improper use can potentially impair the reconstruction results. To illustrate this, we utilize intermediate results from layer 16 of ResNet-18 to reconstruct the user input shown in Figure 14 (a), while providing the LDM with a text prior "yellow pickup park along the road" for feature inversion. Surprisingly, this does not lead to improved reconstruction quality, as depicted in Figure 14 (c). One possible explanation is this description fails to accurately characterize the object within the victim image, resulting in synthetic images that incorrectly represent the foreground in terms of shape, texture and position, as seen in Figure 14 (b). This misalignment further degrades inversion quality. In general, we notice that offering a simple textual description of the image background tends to enhance the reconstruction performance. These descriptions provides an overview on the background of the image, outlining key attributes like the dominant color and surroundings. Our research marks the first step in investigating how textual descriptions influence feature inversion quality. Further investigation is needed to fully grasp the impact of textual priors.

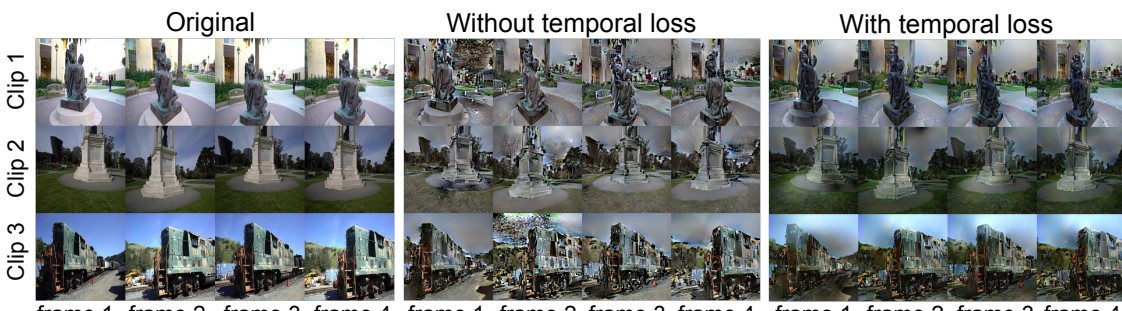

Figure 18: Feature inversion across multiple frames. Our method obtains average increase of 1.8, 7.3 and 0.39 for IS, PSNR and SSIM.

Table 3: Feature inversion quality with and without textual priors. Left/right numbers show results without/with priors.

| Metrics | ResNet-18 (L16) | ResNet-50 (L48) | ViT-base (L8) |
|---------|-----------------|-----------------|---------------|
| IS | 0.33/3.60 | 0.35/3.84 | 0.47/3.05 |
| PSNR | 4.5/15.4 | 5.2/14.9 | 4.3/14.7 |
| SSIM | 0.02/0.59 | 0.03/0.54 | 0.02/0.56 |

### 6.7 Multi-frame Feature Inversion

In this section, we evaluate the multi-frame feature inversion algorithm detailed in Section 4.2. To create multi-frame inputs with high correlations, we utilize the tanks and temples dataset Knapitsch et al. (2017), which contains high-resolution video clips for twelve different objects. We select ten video clips and extract four consecutive frames from each video clip, the time interval between a pair of consecutive two frames is 0.5 seconds. Subsequently, we employ the loss function defined in Equation 10 to jointly reconstruct the four frames. We configure $\lambda_s$ and $\lambda_c$ in Equation 10 to be 1 and 5, respectively. To accelerate the reconstruction process, we set the sampling steps to 10. All other settings remain consistent with the earlier sections.

To demonstrate the advance of the proposed loss function, we conduct a comparison of reconstructed image quality with and without the inclusion of the smoothing loss $\mathcal{L}_c$. From the results presented in Figure 18, we observe a noticeably better reconstruction quality by using the smoothing loss. We also observe that the inclusion of the smoothing loss results in improvements in IS, PSNR, and SSIM scores, with an average increase of 1.8 for IS, 7.3 dB for PSNR, and 0.39 for SSIM across all the video clips.

### 6.8 Ablation Study on Number of Diffusion Sampling Steps

In this section, we investigate how the number of sampling steps affects the reconstruction quality. We utilize the intermediate features from layer 36 of ResNet-50 across 100 inputs from the ImageNet. Subsequently, we conduct feature inversion, as outlined in Section 3.1, employing various sampling step values for the LDM. Table 4 depicts how the IS and SSIM scores evolve with different sampling steps. We observe that both scores increase as the number of sampling steps increases. Nevertheless, both stabilize when sampling steps exceeds 20. Therefore, in this study, we employ a sampling step value of 20 to achieve the optimal balance between feature inversion quality and training efficiency.

Table 4: IS and SSIM scores with different sampling steps.

| Metrics | 10 steps | 15 steps | 20 steps | 25 steps | 30 steps |
|---------|----------|----------|----------|----------|----------|
| IS | 5.93 | 6.38 | 6.55 | 6.62 | 6.65 |
| SSIM | 0.80 | 0.87 | 0.88 | 0.89 | 0.89 |

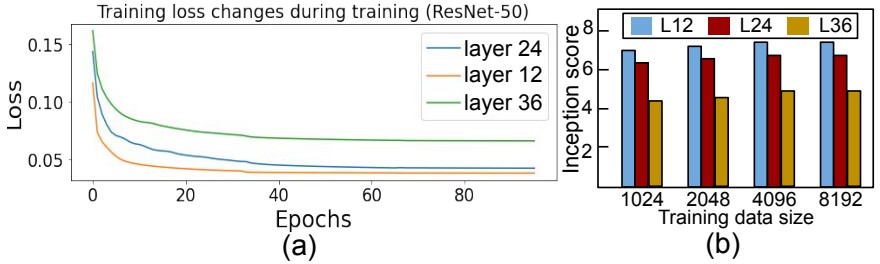

Figure 19: (a) Training loss vs. number of epochs for black-box feature inversion over ResNet-50 on ImageNet. Features are extracted from the end of layer 12, layer 24 and layer 36 respectively. (b) Inception scores with different training data size on ResNet-50 with ImageNet.

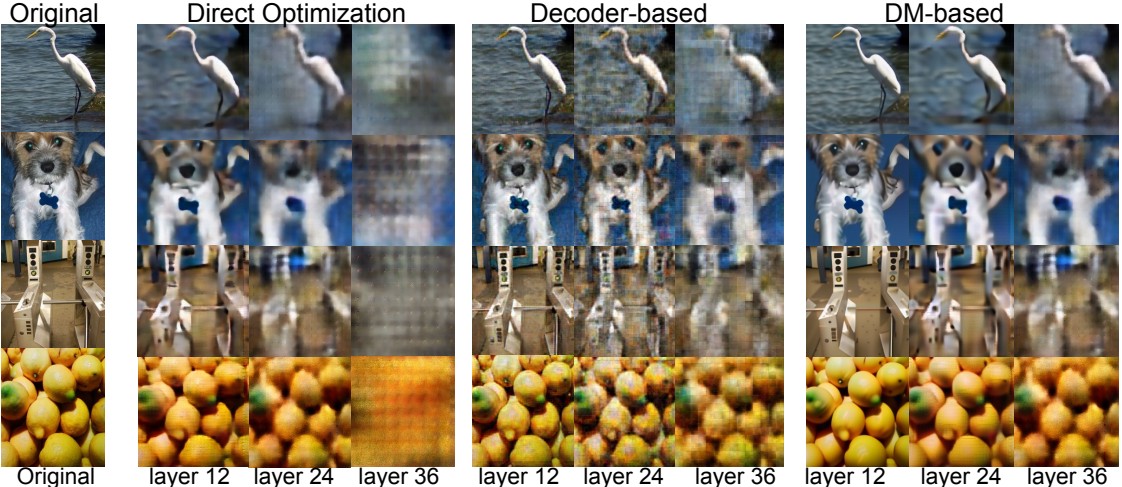

Figure 20: Black-box feature inversion over ResNet-50 on ImageNet, features are extracted from the end of layer 12, layer 24 and layer 36 respectively.

## 6.9 Ablation Study on Loss Function Setting

We conduct an ablation study to evaluate the impact of the loss function components on image reconstruction quality. Specifically, we varied the importance weights $\lambda_s$ and $\lambda_{txt}$ in Equations 9 to assess their influence. The evaluation was carried out by measuring changes in reconstruction quality under white-box inversion attacks on ResNet-50, the results are shown in Table 5. We observe that setting $\lambda_{txt} = 0$, thereby removing the influence of the textual prompt, leads to a decline in reconstructed image quality as the layer depth increases. Additionally, removing the TV loss consistently degrades feature inversion quality across all layer depths.

Table 5: Ablation study on loss function setting.

| Setting | L12 | L24 | L36 |
|---|---|---|---|
| Default | 7.36 | 6.90 | 6.55 |
| Set $\lambda_s = 0$ | 7.11 | 6.47 | 6.20 |
| Set $\lambda_{txt} = 0$ | 7.34 | 6.72 | 5.88 |

## 7 Results for Black-box Feature Inversion

### 7.1 Experiment settings

**Datasets and models:** We use the same datasets, target models and LDM as described in the Section 6.1. To build the training dataset and test dataset of inversion DNN, we randomly select 4096 and 1024 images from the training and test datasets of either ImageNet or YOLO, respectively. We notice that a training data size of 4096 is enough for inversion DNN to generalize well.

Table 6: Evaluation results for black-box inversion.

| Metric | Method | ResNet-18 | | | ResNet-50 | | | ViT-base | | |
|--------|--------|------|------|------|------|------|------|------|------|------|
| | | L4 | L8 | L12 | L12 | L24 | L36 | L3 | L4 | L5 |
| IS | DO | 5.38 | 3.64 | 1.01 | 5.36 | 3.41 | 0.98 | 5.31 | 3.77 | 1.48 |
| | DB | 6.53 | 5.23 | 3.60 | 6.79 | 5.20 | 3.21 | 6.62 | 5.16 | 3.51 |
| | DMB | 6.99 | 6.21 | 5.19 | 7.08 | 6.44 | 4.89 | 6.98 | 6.27 | 5.00 |
| PSNR | DO | 27.6 | 11.1 | 7.74 | 26.3 | 13.1 | 7.11 | 26.6 | 13.2 | 7.80 |
| | DB | 34.3 | 24.3 | 9.6 | 33.4 | 27.0 | 12.3 | 35.3 | 23.9 | 11.5 |
| | DMB | 40.4 | 32.5 | 20.6 | 39.2 | 31.4 | 23.9 | 41.6 | 31.5 | 19.5 |
| SSIM | DO | 0.84 | 0.40 | 0.08 | 0.84 | 0.54 | 0.07 | 0.85 | 0.62 | 0.08 |
| | DB | 0.90 | 0.78 | 0.38 | 0.92 | 0.79 | 0.43 | 0.91 | 0.80 | 0.55 |
| | DMB | 0.92 | 0.84 | 0.46 | 0.94 | 0.88 | 0.67 | 0.95 | 0.88 | 0.71 |

**Hyperparameters:** The inversion DNNs are trained over 96 epochs using a batch size of 128. We assign $\lambda_s$ values of 1 in Equations 11, 12, and 13. We employ the Adam optimizer with an initial learning rate of 0.1 and $\beta$ values of (0.9, 0.999).

**Baseline:** Following the baseline setups outlined in 6.1, we examine the black-box versions of DO and DB. In the case of DO (Figure 17 (a)), we modify the architecture of the inversion DNN to directly reconstruct the user input $x$ without relying on the LDM. Conversely, for DB (Figure 17 (b)), we integrate the decoder from the LDM into the inversion DNN to improve the quality of reconstruction. We change the structure of the inversion DNNs for DMB, DO and DB to ensure they contain an equal number of parameters. DO has been applied by the He et al. (2019); Dong et al. (2021); Maeng et al. (2022) for feature inversion attack under the black-box setting, but DB has not been studied in prior works.

## 7.2 Feature Inversion of Split Models for Image Classification

Similar to Section 6.2, we begin by evaluating the reconstruction accuracy of our black-box feature inversion attacks *without* utilizing text priors. For every target DNN model, we partition them into blocks of layers and capture intermediate outputs at the conclusion of each block. Next, we apply the methods presented in Section 5 to reconstruct the user input. A sample training loss curve for inversion DNN is shown in Figure 19 (a).

Figure 20 illustrates the feature inversion outcomes for ResNet-50 on ImageNet, while Table 6 presents the mean IS, PSNR, and SSIM values computed across the test dataset for different model architectures and feature inversion techniques. Notably, our approach (DM-based) consistently exhibits a higher reconstruction quality across all datasets and DNN architectures. Furthermore, it is worth noting that, under identical settings, the reconstruction quality tends to be lower for the black-box feature inversion attacks compared to the white-box feature inversion attacks, as shown in Section 6.2.

## 7.3 Comparison with Black-box GAN-based Inversion Method

We conduct an additional experiment by replacing the diffusion model with StyleGAN3 under the black-box setting, while keeping all training configurations consistent with those outlined in Section 7.1. The results, presented in Table 7, demonstrate that our diffusion model-based approach achieves superior performance compared to the GAN-based alternative.

## 7.4 Additional Results over YOLO and CLIP

We present additional results using a YOLO-v2 model based on ResNet-50 for object detection. The inputs are reconstructed by the inversion DNN using the intermediate results from layer 12, 24, and 36 (Figure 21 (a)). DMB achieves inception scores of 7.54, 6.99, and 6.80 using features from layers 12, 24, and 36 over the

Table 7: Evaluation results with black-box setting. "StyleGAN-based" and "DMB" denote StyleGAN-based and DM-based approaches, respectively. "L" denotes the layer where the features are extracted. PSNR is shown in db. For IS, PSNR and SSIM, higher values indicate better results.

| Metric | Method | ResNet-18 | | | ResNet-50 | | | ViT-base | | |
|--------|--------|------|------|------|------|------|------|------|------|------|
| | | L4 | L8 | L12 | L12 | L24 | L36 | L3 | L4 | L5 |
| IS | StyleGAN-based | 6.70 | 5.29 | 4.04 | 6.87 | 5.87 | 4.02 | 6.77 | 5.79 | 4.23 |
| | DMB | 6.99 | 6.41 | 5.19 | 7.08 | 6.64 | 4.89 | 6.98 | 6.37 | 5.00 |
| PSNR | StyleGAN-based | 36.8 | 27.6 | 11.4 | 35.4 | 28.9 | 16.7 | 37.0 | 26.4 | 14.8 |
| | DMB | 40.4 | 32.5 | 20.6 | 39.2 | 31.4 | 23.9 | 41.6 | 31.5 | 19.5 |
| SSIM | StyleGAN-based | 0.92 | 0.80 | 0.41 | 0.93 | 0.82 | 0.54 | 0.93 | 0.82 | 0.60 |
| | DMB | 0.92 | 0.84 | 0.46 | 0.94 | 0.88 | 0.67 | 0.95 | 0.88 | 0.71 |

Table 8: Black-box feature inversion results with/without textual priors. Left/right numbers show results without/with priors.

| Metrics | ResNet-50 (L48) | YOLO (L48) | CLIP (End-to-end) |
|---------|-----------------|------------|-------------------|
| IS | 0.29/3.11 | 0.88/3.48 | 3.22/4.09 |
| PSNR | 4.3/13.6 | 5.4/14.0 | 12.0/17.7 |
| SSIM | 0.03/0.50 | 0.08/0.44 | 0.46/0.56 |

test dataset of COCO. This is much higher than IS scores obtained by DO method (5.64, 3.97 and 1.20 for L12, L24 and L36, respectively) and DB method (6.95, 5.95 and 4.46 for L12, L24 and L36, respectively).

Moreover, in Figure 21 (b), we showcase the outcomes of black-box end-to-end feature inversion using the CLIP Radford et al. (2021) image encoder. Specifically, we curate a subset of 1024 images from the ImageNet test dataset and achieve an IS, PSNR, and SSIM of 3.22, 12.0, and 0.46, respectively. Contrasting this, the DO method yields an IS, PSNR, and SSIM of 0.45, 2.62, and 0.09, respectively, while the DB method returns IS, PSNR, and SSIM scores of 2.03, 9.10, and 0.25, correspondingly. Similarly, we observe that CLIP is notably more amenable to inversion at the same layer depth compared to a DNN tailored for image classification.

### 7.5 Impact of Text Prior on Feature Inversion

In this section, we analyze the impact of textual prior on the inversion results. Especially, we extract the intermediate feature from layer 16 in ResNet-18 and layer 48 from YOLO, we extract intermediate features from deeper layers, such as layer 48 in ResNet-50, and invert the image with simple background, as described in Section 6.6. $\lambda_s$ and $\lambda_{txt}$ in Equation 9 are set to 1 and 3, respectively.

Table 8 highlights the quantitative results for feature inversion with and without textual prior in terms of average IS, PSNR and SSIM. Evidently, including the textual prior significantly improves reconstruction quality quantitatively as well. However, we also observe that a similar failure case as described in Section 6.6 under black-box setting.

### 7.6 Multi-frame Feature Inversion

In this section, we evaluate the multi-frame feature inversion algorithm detailed in Section 10.9. We employ the same training and testing datasets as described in 6.7. Specifically, the training and test datasets include 1024 and 256 video clips, respectively. $\lambda_s$ in Equation 13 are set to 1. We conduct a comparison of reconstructed image quality with and without the inclusion of the pointwise convolution layer described in Figure 9, whose primary function is to consider the temporal correlation during the reconstruction of input frames. Moreover, we adjust the structure of the DNN so that the total amount of parameters are the same for both scenarios. We note a significant enhancement in IS, PSNR, and SSIM scores with the inclusion of

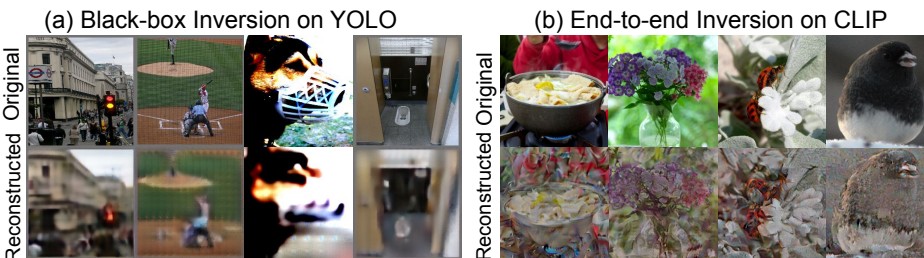

Figure 21: (a) Black-box feature inversion using the output from layer 36 of YOLO. (b) End-to-end black-box feature inversion attack over CLIP.

the pointwise convolutional layer, resulting in an average increase of 0.2 for IS, 4.0 dB for PSNR, and 0.09 for SSIM across all video clips in the test dataset using the intermediate results from layer 36 of ResNet-50 on ImageNet dataset. More evaluation results are presented in the appendix.

### 7.7 Ablation Study on Training Set Size

We study the impact of training dataset size on reconstruction quality. We vary the size of the ImageNet training dataset while maintaining the test dataset size at 1024, and measure feature inversion outcomes using the black-box setting. Figure 19 (b) demonstrates that IS scores steadily increase as the training dataset size grows. Interestingly, even with a batch size of 1024, a notably high IS score is achieved, suggesting that a smaller training dataset can still facilitate effective generalization of the inversion DNN.

## 8 Discussion

In this section, we summarize some findings we observe from the evaluation results (Section 8.1). We then discuss the potential defense strategies in Section 8.2.

### 8.1 Insights From the Evaluation Results

**A deeper DNN does not guarantee privacy:** Based on the results outlined in Section 6.2 and Section 7.2, it becomes evident that a deeper DNN does not inherently ensure privacy. For instance, as observed in Table 1 and Table 6, the quality of reconstructed input using the intermediate features at the 12th layer of ResNet-18 is notably inferior to that of the 24th layer output of ResNet-50. This suggests that the absolute layer depth alone does not guarantee any privacy protection. Instead, what matters is the **relative layer depth** within the DNN. For instance, the reconstructed quality using the outputs of a middle layer of ResNet-50 (e.g., 24) is approximately equal to that from the middle layer (e.g., 8) of ResNet-18.

**Transformer is harder to invert than CNN:** Another trend we notice is that transformers, such as ViT, exhibit better privacy protection capabilities compared to Convolutional Neural Networks (CNNs). As indicated by the results in Table 1 and Table 6, the quality of reconstructed input using features from the middle layer (e.g., 5) of ViT-base is comparable to that obtained using features from later layers in ResNet-18 or ResNet-50. This may contribute to the fact that transformers with self-attention mechanism inherently amounts to a low-pass filter Wang et al. (2022a), this will eliminate a lot of high-frequency information within the original input image, making the intermediate features harder to invert. By contrast, CNNs typically extract information across wide frequency ranges Yosinski et al. (2015), thereby retaining more essential information for feature inversion. Nevertheless, further studies are needed.

**Self-supervised pretrained backbone models are easier to invert:** The evaluation results shown in Section 6.5 and Section 7.4 illustrate that pretrained backbone models with self-supervised learning, are more amenable to inversion compared to DNNs trained with supervised learning frameworks tailored for specific tasks like image classification. This is due to the fact that the SSL-pretrained backbones tend to preserve a rich set of information that can be beneficial for various downstream tasks. The pretraining process typically

involves learning representations that capture meaningful patterns and structures in the input data, which can generalize well to different tasks. In contrast, DNNs trained with supervised learning using labeled datasets tend to eliminate redundant information unrelated to the task during the training process, making the input data more challenging to reconstruct.

**A well-constructed textual prompt can improve inversion performance:** This work is the first work to demonstrate that additional information in another format (i.e., textual format) can be utilized to enhance reconstruction performance. Our general finding is that a simple textual description consisting of a few words that provides an overview of the image background, highlighting attributes like the dominant color and surroundings, can generally enhance reconstruction quality. Although applying an inaccurate textual prior will degrade the attack quality, if the attacker has multiple candidate textual descriptions, the best strategy is to exhaustively try all of them and select the one that achieves the optimal quality. While using textual priors in feature inversion is not our main focus, it is a promising area for future research.

In real-world scenarios, descriptive information about the victim's input can often be inferred or leaked. In split computing settings, attackers may gain access to contextual prior knowledge that enables them to craft effective textual prompts, enhancing the success of feature inversion attacks. For example, in smart home or AR/VR environments, common indoor contexts such as a living room or kitchen are easily inferred. In professional applications like Codec Avatars, an attacker may know that the user is participating in a virtual meeting. Location-based AR applications can reveal contextual clues through GPS, allowing prompts such as "Eiffel Tower on a clear day." Similarly, in autonomous driving or fitness apps, route history or usage data may suggest likely scenes such as "a suburban street" or "a person doing yoga indoors." These contextual cues, obtained through metadata, environmental knowledge, or user behavior, make prompt-based inversion attacks both practical and more effective.

**Revealing the model weight and structure can improve the quality of the attack:** Based on the evaluation results presented in Table 1 and Table 6, it is evident that the feature inversion attack achieves higher quality results in the white-box setting compared to the black-box setting. This underscores the importance of weight and architecture of the user model in influencing the quality of feature inversion.

## 8.2 Defense over Feature Inversion Attack

In this section, we explore potential defense strategies to mitigate the feature inversion attacks outlined in Section 4 and Section 5. One defensive strategy is to ensure that all DNN computation and communication happen over encrypted data, i.e. through cryptographic methods like secure multiparty computation (MPC) or homomorphic encryption (HE).

Secure MPC allows a group of $n$ untrusting parties to collaboratively compute a public function $f(x_1, x_2, ..., x_n)$ over their private inputs $x_1, x_2, ..., x_n$ without revealing any of their secret information. If the MPC scheme is expressive enough to implement large and complex neural networks like diffusion models, the implementation often has prohibitively large communication overheads and high computational complexity. For example, recent work on MPC implementation of VGG16 in the WAN setting leads to 37s latency (and training can take several weeks) Wagh et al. (2020).

Similarly, HE schemes allow certain operations, such as arithmetic or boolean functions, to be applied to the ciphertext, thereby allowing privacy-preserving neural network evaluations without revealing sensitive information in plaintext form. But computing on ciphertexts over plaintext means both higher communication and computation costs. Additionally, HE requires additional computation steps like noise-management via bootstrapping. Prior implementations show significant slowdown, 300s for encryption, DNN application, and decryption (up to 30s for a 5x5 convolutional layer to a simple 5-layer MNIST network, and 127s for pooling) Gilad-Bachrach et al. (2016). While these are promising and active research areas, at this time these methods are not widely deployed due to large overheads over an already computationally intense DNN architecture.

Another feasible approach involves the use of differential privacy (DP) to introduce noise and obscure sensitive information. In particular, random noise $\epsilon$ can be directly integrated into the intermediate results $z_{mid}$, with the magnitude of the noise being controlled to achieve the desired level of DP. Nevertheless, it's crucial to

acknowledge the trade-off between usability and privacy. When higher levels of noise are introduced, there will be a corresponding drop in model accuracy. To mitigate this loss in accuracy, it is beneficial to take into account the influence of the injected noise during the training phase for the target DNN $F_\theta(.)$.

## 9 Conclusion and Future Work

In this study, we demonstrate the significant performance enhancement achievable in the feature inversion process via the utilization of the diffusion model. We also highlight the potential for utilizing diverse forms of prior knowledge, such as textual information and cross-frame correlations, to further improve the reconstruction quality. From the evaluation results, we show that GenAI, with its remarkable ability to synthesize realistic and coherent data, can also be utilized to detrimentally affect individuals' lives, particularly in the context of privacy breaches. This opens up interesting future avenues in a promising direction of research.

Several directions remain for future work. First, although textual input has been shown to enhance image reconstruction, an open and compelling question is how to automatically generate prompts with the optimal level of detail to maximize reconstruction quality. Second, given the rapid advances in generative image models, it would be valuable to investigate the effectiveness of newer architectures, such as diffusion transformers Peebles & Xie (2023). Finally, exploring feature inversion attacks in the context of foundation models presents an important and promising area for further study.

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

## 10 Appendix

### 10.1 statement on Data Availability

Due to institutional restrictions, we are unable to use the public latent diffusion model for publishing our research outcomes. Instead, we employed an LDM with an architecture highly similar to Stable Diffusion 2.1 in terms of architecture, model size and pretraining techniques. Our internal model was pretrained on the dataset collected by a third party (Shutterstock) that is not public. Regarding the dataset, it consists of 385 million images: 321 million without people and 64 million with people. In addition, we have also previously conducted extensive experiments with the public Stable Diffusion 2.1, which yielded similar (and even better) results than the reported results in terms of IS, PSNR, and SSIM scores. If the paper is accepted, we intend to release the code as open source. This code will enable the integration of the public LDM for conducting feature inversion attacks.

### 10.2 Implementation details

Table 9 and Table 10 list the detailed settings for feature inversion described in Section 4 and Section 5. To initiate the training process, the latent variable v is initialized using a randomly generated vector sampled from a normal distribution with a standard deviation of 0.1.

| General Configuration | Detail |
|---|---|
| Optimizer | Adam |
| Total iterations | 1500 |
| Base learning rate | 0.1 |
| Learning rate schedule | multiple stages |
| Sampling steps | 20 |
| $\lambda_s$ | 1 |

Table 9: Detailed settings for white-box feature inversion.

| General Configuration | Detail |
|---|---|
| Optimizer | Adam |
| Total epochs | 96 |
| Base learning rate | 0.1 |
| Learning rate schedule | multiple stages |
| Sampling steps | 20 |
| Batch size | 128 |
| Training data size | 4096 |
| Test data size | 1024 |

Table 10: Detailed settings for black-box feature inversion.

In Figure 23, we illustrate the changes on the loss function throughout the reconstruction process as outlined in Algorithm 1 using the intermediate features from ResNet-50. It is evident that the loss values converge by the end of the 1500 iterations.

### 10.3 Feature inversion training with textual prior

Algorithm 3 describes the algorithm for feature inversion training with text prior under white-box settings.

---

**Algorithm 3** Feature Inversion with Text Prior

---

$F_1(.)$ is the user DNN model.
$v^m$ is the input latent vector of LDMs at iteration $m$.
$M$ is total number of iterations.
$\epsilon$ is the learning rate.
$t_{prior}$ is the prior knowledge described in text.
$E()$ is the pretrained text encoder.

**for** $1 \leqslant m \leqslant M$ **do** $v_n^m = \frac{v^m - \text{mean}(v^m)}{\text{std}(v^m)}$
$\mathcal{L}_{tot} = ||F_1(D(v_n^m, E(t_{prior}))) - z_{mid}||^2 + \lambda_s TV(D(v_n^m)) + \lambda_{txt}||z_n - q||^2$
$v^{m+1} = v^m - \epsilon \frac{d\mathcal{L}_{tot}}{dv}$
$m = m + 1$
$v_n = \frac{v^M - \text{mean}(v^M)}{\text{std}(v^M)}$
**return** $D(v_n^M, E(t_{prior}))$.

---

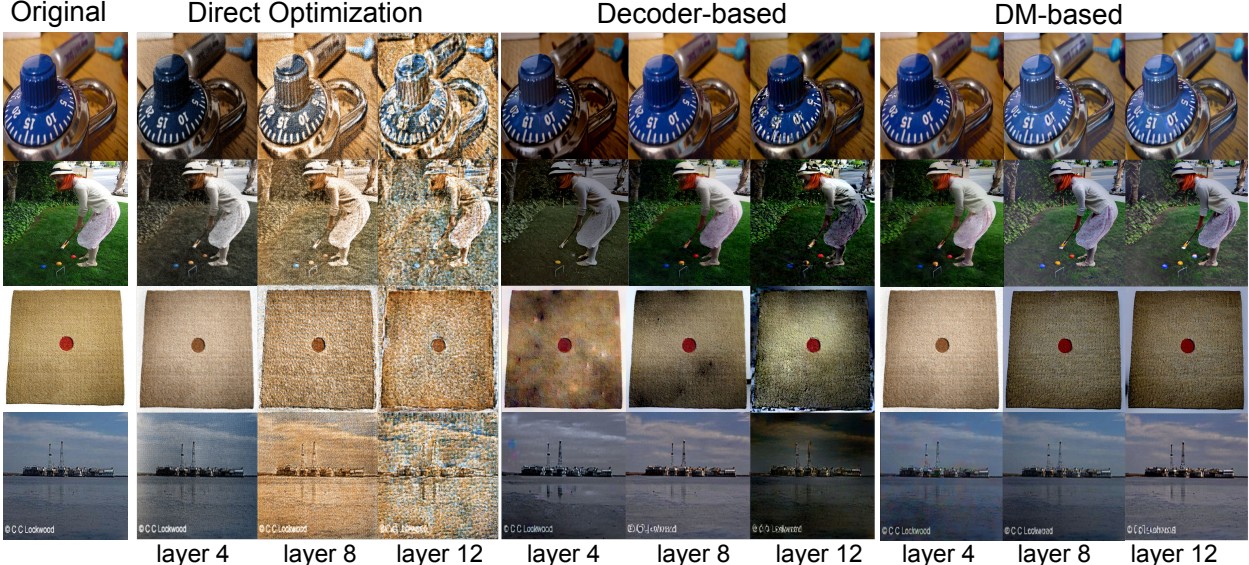

Figure 22: Feature inversion of ResNet-18 on ImageNet with white-box setting.

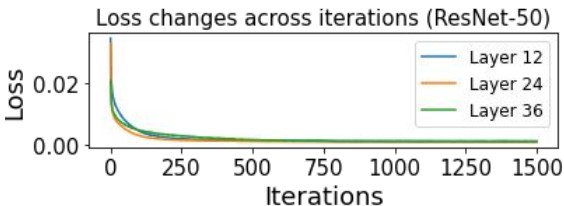

Figure 23: Changes on training loss during the inversion process.

## 10.4 Multi-frame feature inversion training

Algorithm 2 describes the algorithm for feature inversion training with multiple frames under white-box settings.

## 10.5 Training algorithm for feature inversion under black-box settings

Algorithm 4 describes the algorithm for feature inversion training with black-box setting.

---
**Algorithm 4** Black-box Feature Inversion
---
$F_1(.)$ is the user DNN model.
$X_q = \{x_q\}$, $Y_q = \{y_q\}$ are the sets training input samples and the corresponding intermediate results from the user DNN $F_1(.)$.
$(x_q, y_q)$ is a training sample. $\lambda_s$ is the weight for the TV loss.
$T$ is total number of iterations.
$\epsilon$ is the learning rate.
$F_\theta^{inv}(.)$ is the inversion DNN.
$D(.)$ is the latent diffusion model.
Initialize $\theta$ within $F_\theta^{inv}(.)$.

**for** $1 \leqslant e \leqslant E$ **do**
    **for** $(x_q, y_q) \in (X_q, Y_q)$ **do** $z = F_\theta^{inv}(y_q)$
$x = D(z_q)$
$\mathcal{L}_{tot} = ||x - x_q||^2 + \lambda_s TV(x)$
Compute the gradient and update $\theta$.
**return** $\theta$.

---

## 10.6 More results on white-box feature inversion

Figure 22 shows the feature inversion results for ResNet-18 on ImageNet dataset. Finally, Figure 24 shows the feature inversion results for ViT on ImageNet.

## 10.7 More results on black-box feature inversion

Figure 25 illustrate the feature inversion outcomes for ViT on ImageNet.

## 10.8 Impact of sampling steps

In this section, we show the impact of the DM sampling steps on the feature inversion results for white-box setting. Specifically, we show the reconstructed input images (Figure 26) from layer 36 of ResNet-50 on ImageNet under white-box settings. This serves as a supplementary addition to the ablation studies discussed in Section 6.8. We observe that the feature inversion quality improves as the number of sampling steps increasing from 10 to 20.

+

Table 11: Multi-frame feature inversion results under black-box settings. For ResNet-50 and ViT, features are inverted using the outputs from L36 and L5, respectively. The number on the left/right represents the results obtained without/with a pointwise convolutional layer.

| Metrics | IS | PSNR | SSIM |
|---------|-----|------|------|
| ResNet-50 | 4.82/4.99 | 23.6/27.2 | 0.67/0.76 |
| ViT | 4.89/5.10 | 19.8/22.2 | 0.70/0.77 |

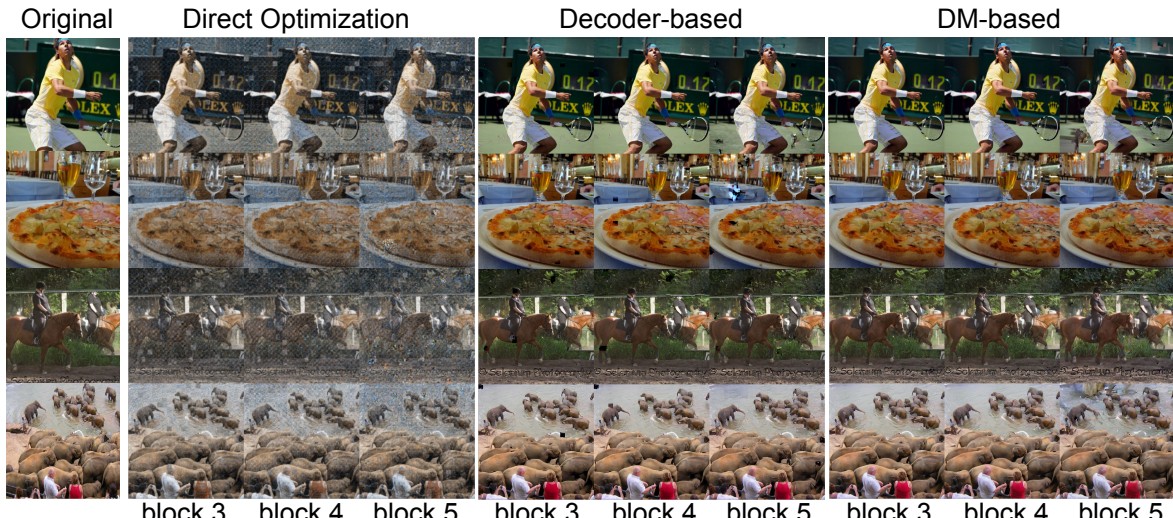

Figure 24: Feature inversion of ViT on ImageNet with white-box setting.

## 10.9 Multi-frame inversion with black-box settings

In this section, we show the multi-frame feature inversion results under black-box settings (Table 11). We evaluate using two target models, ResNet-50 and ViT. For ResNet-50 and ViT, their features are inverted using the intermediate results from L36 and L5, respectively. We can see that involving the pointwise layer in the inversion DNN obtains a clear improvement on the reconstruction quality.

## 10.10 More results on inversion with textual prior

In this section, we show additional results on feature inversion with textual prior (Figure 27) under white-box settings. Specifically, we use the same textual prior as Figure 16. Clearly, we can notice a significant improvement in quality when incorporating a textual prior in feature inversion.

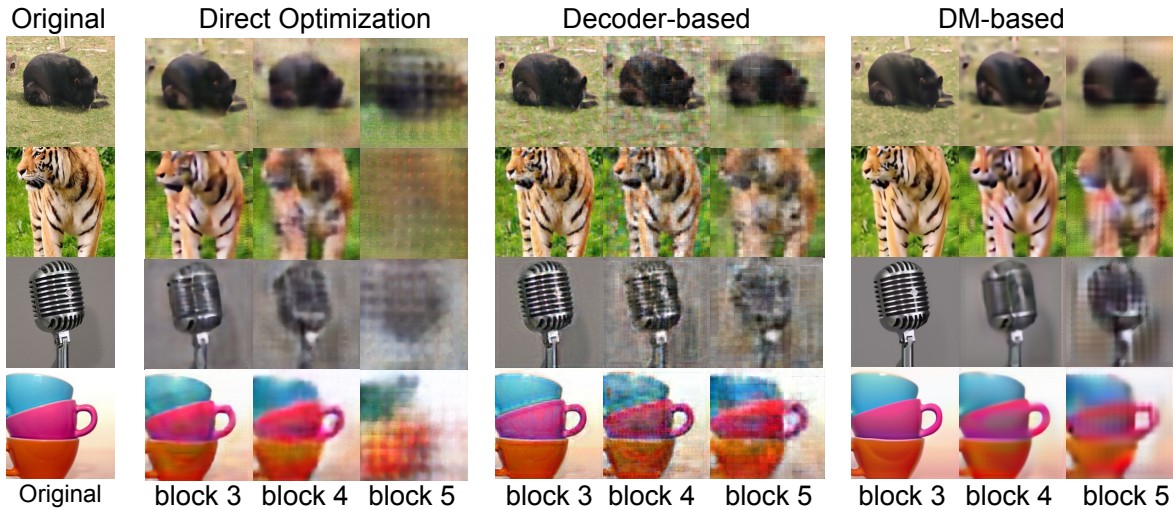

Figure 25: Feature inversion of ViT on ImageNet with black-box setting.

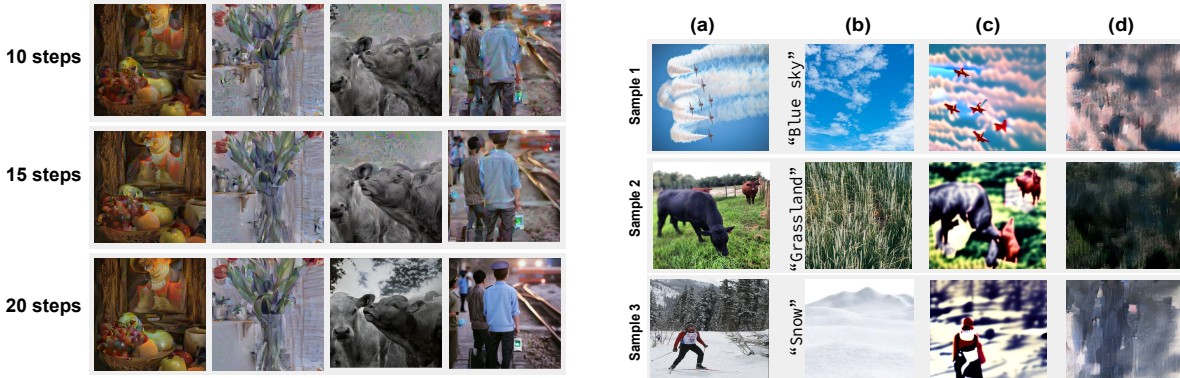

Figure 26: Feature inversion results with different sampling steps.

Figure 27: Feature inversion with textual prior.

