# OpenReview forum: "Unlocking Visual Secrets: Inverting Features with Diffusion Priors for Image Reconstruction"
_TMLR — Accepted by TMLR_

### Review · Reviewer_m7AQ · 2025-03-29

**Summary Of Contributions:**

This paper deals with a privacy attack paradigm called “feature inversion” in which privacy data is leaked via reconstructing the original input from the intermediate features. Specifically, this work focuses on the visual domain and latent-diffusion-based image generation models. The main contributions of this work include improving feature inversion using diffusion models, and extensions that add support for textual priors and inverting features for videos. The reported reconstruction performance surpasses previous methods, leading to a new state-of-the-art.

**Audience:**

Yes

**Broader Impact Concerns:**

I think that given this paper delas with privacy attacks, a "Broader Impact" section would be appropriate.

**Claims And Evidence:**

Yes

**Requested Changes:**

* Question: it seems the pre-trained encoders were all supervised (except for CLIP, which is trained with text-image pairs). What is the authors’ opinion on using pre-trained self-supervised encoders (such as [DINOv2](https://arxiv.org/abs/2304.07193) for example)?
* Please go over all the figures and modify the captions to be more informative.
* See “Weaknesses” and “Minor” above.

**Strengths And Weaknesses:**

**Strengths**:
* Sufficient background.
* The manuscript clearly explains the distinction between white-box and black-box types of attacks.
* In general, the paper is very comprehensive and covers several aspects of attacks.
* Great performance.
* Detailed insights.
* Promise for an open-source code and integration with public models (appendix).

**Weaknesses**:
* Missing ablations of additional losses like the TV loss, “Gaussianity” loss, and the disparity loss. Are they really necessary?
* The authors mention that inverting ViT features is more challenging. Perhaps the combination of ViT features and U-Net-based models is incompatible, and instead Transformer-based models should be used. I appreciate the insights in the conclusions section.
* I find the conclusion in the text-based section (“In general, we notice that offering a simple textual description of the image background tends to enhance the reconstruction performance. These descriptions provides an overview on the background of the image, outlining key attributes like the dominant color and surroundings.”) a bit counter-intuitive, as I would assume the background should not be the important part when trying to extract important information (also “provides” -> “provide”).
* I find the comparison to baselines a bit lacking in my opinion, as there are many several works covered in the related work part of the paper. If it is impossible to compare these baselines, the authors should at least report why. I acknowledge that I’m not an expert in the privacy field and thus cannot provide an expert opinion regarding the right baselines.
* Lacking discussion of limitations and future work.

**Minor**:
* Figures could generally be more detailed and explain what is illustrated. For example, Figures 2-4: very hard to understand what the authors want the reader to observe.
* Page 6, Section 3.3, second paragraph: “by making $F_1 = F$ and $F_2=$,. $F_2=$?
* Algorithm 1: how is $v$ initialized? Also, $\epsilon$ is not a great choice for denoting learning rate in my opinion.
* Equation 6 (page 6) appears out of nowhere. I would suggest building the loss gradually if the discussion of the TV loss comes later.
* Page 10, “Hyperparamters”: all equation references should use a capital E, e.g. “Equation 6”. Similarly on page 16 Section 7.1.
* Page 13, Section 6.3 - what does “We select 100 test datasets from the COCO dataset for inversion” mean?
* Appendix: page 1 - broken reference to “Algorithm”. Similarly in page 2 Section 1.4. Also broken references to sections and figures throughout the rest of the appendix.

---

> ### Author Response · Authors · 2025-06-03
>
> Thank you for the insightful review and feedback.
>
> **Q1:** Missing ablations of additional losses like the TV loss, “Gaussianity” loss, and the disparity loss. Are they really necessary?
>
> **A:** We conducted an ablation study to evaluate the impact of the loss function components on image reconstruction quality. Specifically, we varied the importance weights λs and λtxt in Equations 9 and 12 to assess their influence. The evaluation was carried out by measuring changes in reconstruction quality under both white-box and black-box inversion attacks on ResNet-50.
> | Setting            | L12  | L24  | L36  |
> |--------------------|------|------|------|
> | Default            | 7.36 | 6.90 | 6.55 |
> | Setting λs = 0     | 7.11 | 6.47 | 6.20 |
> | Setting λtxt = 0   | 7.34 | 6.72 | 5.88 |
>
> We observe that setting λtxt = 0, thereby removing the influence of the textual prompt, leads to a decline in reconstructed image quality as the layer depth increases. Additionally, removing the TV loss consistently degrades feature inversion quality across all layer depths.
>
> **Q2**: The authors mention that inverting ViT features is more challenging. Perhaps the combination of ViT features and U-Net-based models is incompatible, and instead Transformer-based models should be used. I appreciate the insights in the conclusions section.
>
> **A**: We will incorporate additional insights into the conclusion section. In addition, Section 8 already provides some analysis explaining why ViT performs worse in terms of feature inversion, which may be attributed to the fact that transformers with self-attention mechanisms inherently behave like low-pass filters, which suppress high-frequency information in the input image. As a result, the intermediate features become more difficult to invert.
>
> **Q3**: I find the conclusion in the text-based section (“In general, we notice that offering a simple textual description of the image background tends to enhance the reconstruction performance. These descriptions provide an overview on the background of the image, outlining key attributes like the dominant color and surroundings.”) a bit counter-intuitive.
>
> **A**: Our general finding is that a brief textual description containing only a few key attributes can significantly enhance image reconstruction quality. However, overly detailed descriptions may constrain the diffusion model too tightly, reducing its generative flexibility and ultimately harming inversion quality. In practice, textual descriptions are rarely perfectly accurate, as shown in Figure 14. For instance, describing the exact position of an object using text is difficult, yet spatial placement plays a crucial role in reconstruction quality. Therefore, using a rough and concise textual description tends to yield better results.

---

> > ### Author Response · Authors · 2025-06-03
> >
> > **Q4**: I find the comparison to baselines a bit lacking in my opinion, as there are many several works covered in the related work part of the paper. If it is impossible to compare these baselines, the authors should at least report why.
> >
> > **A**: While there has been work on gradient inversion and model inversion attacks, to the best of our knowledge, relatively few studies have explored the use of advanced prior models in feature inversion attack. We compared our method with DO and DB by substituting the diffusion model, as described in Sections 5 and 6, where DO and DB were adopted in prior works [1][2]. Additionally, we conducted another experiment by replacing the diffusion model with GAN (StyleGAN3), following a similar approach. Specifically, we modified the pipelines in Figures 4 and 9 by substituting the diffusion model with StyleGAN3 and trained its latent vector to reconstruct the user input. The evaluation was performed under both white-box and black-box settings, with all training configurations kept consistent with those outlined in Sections 6.1 and 7.1. The results are presented in the following tables, corresponding to the white-box and black-box evaluations, respectively. We can see that our diffusion model-based approach outperforms the GAN-based approach.
> >
> > These are the results for the white-box inversion attack:
> > |  | Metric | Method         | L4 | L8   | L12  |
> > |--------|--------|----------------|--------------|------|------|
> > |        | IS     | StyleGAN-based | 7.02         | 6.22 | 5.06 |
> > |        |        | DMB            | 7.23         | 6.86 | 6.48 |
> > |        | PSNR   | StyleGAN-based | 37.3         | 34.1 | 22.5 |
> > |        |        | DMB            | 41.0         | 36.3 | 29.1 |
> > |        | SSIM   | StyleGAN-based | 0.94         | 0.92 | 0.78 |
> > |        |        | DMB            | 0.97         | 0.94 | 0.86 |
> >
> > These are the results for the black-box inversion attack:
> > |        | Metric | Method         | ResNet-18 L4 | L8   | L12  |
> > |--------|--------|----------------|--------------|------|------|
> > |        | IS     | StyleGAN-based | 6.70         | 5.29 | 4.04 |
> > |        |        | DMB            | 6.99         | 6.41 | 5.19 |
> > |        | PSNR   | StyleGAN-based | 36.8         | 27.6 | 11.4 |
> > |        |        | DMB            | 40.4         | 32.5 | 20.6 |
> > |        | SSIM   | StyleGAN-based | 0.92         | 0.80 | 0.41 |
> > |        |        | DMB            | 0.92         | 0.84 | 0.46 |
> >
> > We can see that the results for StyleGAN-based method is not as good as ours.
> >
> > [1] Kiwan Maeng, Chuan Guo, Sanjay Kariyappa, and Edward Suh. Measuring and controlling split layer privacy leakage using fisher information. arXiv preprint arXiv:2209.10119, 2022.
> > [2] Zecheng He, Tianwei Zhang, and Ruby B Lee. Model inversion attacks against collaborative inference. In Proceedings of the 35th Annual Computer Security Applications Conference, pp. 148–162, 2019.
> >
> > **Q5**: Lacking discussion of limitations and future work.
> > There are several limitations in the current scheme. First, while textual input has been shown to effectively enhance image reconstruction, determining how to automatically generate prompts with the optimal level of detail for best reconstruction quality remains an open and interesting question. Second, with the rapid progress in generative image models, it would be valuable to explore the use of more recent architectures such as diffusion transformers and VAR for further evaluation. Finally, it will be interesting to study the feature inversion attack over the foundation models. We will add it in the final version of the work.
> >
> > **Q6**:  Question: it seems the pre-trained encoders were all supervised (except for CLIP, which is trained with text-image pairs). What is the authors’ opinion on using pre-trained self-supervised encoders (such as DINOv2 for example)?
> >
> > **A**: Thanks for bringing this up. We discuss this in Section 8. Overall, our findings suggest that self-supervised pretrained backbone models are more susceptible to inversion. As demonstrated in the evaluation results in Sections 6.4 and 7.3, these models are generally easier to invert compared to those trained with supervised learning for specific tasks such as image classification. Additional discussion is provided in Section 8.
> >
> > **Q7**: Algorithm 1: how is v initialized?
> >
> > **A**: v is initialized with values drawn from a standard normal distribution.
> >
> > Thank you for the detailed comments on grammar and writing. We will revise the figure captions to make them more informative in the final version of the paper.

---

> > > ### Comment · Reviewer_m7AQ · 2025-06-07
> > > **Thank you for your effort**
> > >
> > > I thank the authors for their effort during the discussion period and for uploading a revised version. Overall, I think these modifcations improve the paper and I'm happy recommneding accepting it. One point that is still missing in the revised version is the loss functions ablation (which appears in the response for my review), I believe it should be in the paper as well.

---

> > > > ### Author Response · Authors · 2025-06-07
> > > >
> > > > Dear Reviewer, thank you for your positive feedback! We have added the ablation study on the loss function to Section 6.9 of the paper.

---

### Review · Reviewer_pjJm · 2025-04-21

**Summary Of Contributions:**

The paper focuses on using diffusion models to improve image inversion quality from internediate DNN features. The authors formulate the problems into the balck box and the white box settings and propose different algorithms for the improvement. Through comprehensive ablation studies and analysis, the authors demonstrate incorporating various forms of prior knowledge, such as textual prompts and cross-frame temporal correlations, to improve the quality of inverted features with DMs.

**Audience:**

Yes

**Claims And Evidence:**

Yes

**Requested Changes:**

Needs clarification of the weakness and questions in the section above.

**Strengths And Weaknesses:**

## Strengths
1. The ablation studies are comprehensive, where the authors include different baselines like resnet, vit, clip, yolo with different layers/depth to prove the soundsness of the proposed approach.
2. Both qualatitive and quantative results looks good.
3. The writing is good easy to follow.

## Weakness
1. The authors should compare with other baselins directly (instead of similar model configurations) to better prove the effectiveness of th proposed methods.
2. DMS, thought effective, could face latency and memory problems in real world applications. The authors should have further discussions in this issue.
3. It seems text prioirs have huge impact on the final results, then the accuracy of the description will be a bottleneck, otherwise the authors need to have experiments for further discussion.

## Questions
1. Elaborate more about the difference between black and white box setting. In 3.2, it mentions,  "the key distinction being the relaxation of the assumption regarding the adversary’s knowledge of F1(.). Here, the adversary can only gather information about F1 indirectly through querying it." However, eq.4 in the black box setting can also be achieved by simply querying F1 w/o knowing its architecture?
2. 3.3 typo, F2=?

---

> ### Author Response · Authors · 2025-06-03
>
> Thank you for the insightful review and feedback.
>
> **Q1:**  The authors should compare with other baselines directly (instead of similar model configurations) to better prove the effectiveness of the proposed methods.
>
> **A:** We compared our method with DO and DB by substituting the diffusion model, as described in Sections 5 and 6, where DO and DB were adopted in prior works [1][2]. Additionally, we conducted another experiment by replacing the diffusion model with GAN (StyleGAN3), following a similar approach. Specifically, we modified the pipelines in Figures 4 and 9 by substituting the diffusion model with StyleGAN3 and trained its latent vector to reconstruct the user input. The evaluation was performed under both white-box and black-box settings, with all training configurations kept consistent with those outlined in Sections 6.1 and 7.1. The results are presented in the following tables, corresponding to the white-box and black-box evaluations, respectively. We can see that our diffusion model-based approach outperforms the GAN-based approach.
> These are the results for the white-box inversion attack:
> |  | Metric | Method         | L4 | L8   | L12  |
> |--------|--------|----------------|--------------|------|------|
> |        | IS     | StyleGAN-based | 7.02         | 6.22 | 5.06 |
> |        |        | DMB            | 7.23         | 6.86 | 6.48 |
> |        | PSNR   | StyleGAN-based | 37.3         | 34.1 | 22.5 |
> |        |        | DMB            | 41.0         | 36.3 | 29.1 |
> |        | SSIM   | StyleGAN-based | 0.94         | 0.92 | 0.78 |
> |        |        | DMB            | 0.97         | 0.94 | 0.86 |
>
> These are the results for the black-box inversion attack:
> |        | Metric | Method         | ResNet-18 L4 | L8   | L12  |
> |--------|--------|----------------|--------------|------|------|
> |        | IS     | StyleGAN-based | 6.70         | 5.29 | 4.04 |
> |        |        | DMB            | 6.99         | 6.41 | 5.19 |
> |        | PSNR   | StyleGAN-based | 36.8         | 27.6 | 11.4 |
> |        |        | DMB            | 40.4         | 32.5 | 20.6 |
> |        | SSIM   | StyleGAN-based | 0.92         | 0.80 | 0.41 |
> |        |        | DMB            | 0.92         | 0.84 | 0.46 |
>
> We can see that the results for StyleGAN-based method is not as good as ours.
>
> **Q2:** DMS, thought effective, could face latency and memory problems in real world applications. The authors should have further discussions in this issue.
>
> **A:** Typically, attackers aim to reconstruct high-quality input images, a process that does not require real-time execution. Furthermore, these attacks are carried out on the attacker's own machines, which are often far more powerful than edge devices. Consequently, such attacks are not limited by the computational constraints of split computing and are not executed on resource-constrained edge platforms.
>
> **Q3:** It seems text priors have a huge impact on the final results, then the accuracy of the description will be a bottleneck, otherwise the authors need to have experiments for further discussion.
>
> **A:** Our general finding is that a brief textual description containing just a few key attributes can significantly enhance image reconstruction quality. However, overly detailed descriptions may overly constrain the diffusion model, limiting its generative flexibility and ultimately degrading inversion quality. Additionally, in practice, textual descriptions are rarely perfectly precise, as illustrated in Figure 14. For example, accurately conveying the exact position of an object using text alone is challenging, yet spatial positioning has a significant impact on the quality of the reconstructed image. Therefore a rough textual description is better.
>
> **Q4:** Elaborate more about the difference between black and white box setting. In 3.2, it mentions, "the key distinction being the relaxation of the assumption regarding the adversary’s knowledge of F1(.). Here, the adversary can only gather information about F1 indirectly through querying it." However, eq.4 in the black box setting can also be achieved by simply querying F1 w/o knowing its architecture?
>
> **A:** That’s correct, for a black-box attack, the attacker only needs access to the function F1(.) by querying it with their own inputs and collecting the outputs for training. The internal architecture of F1 does not need to be known. We will clarifying it in the final version of the paper.
>
> **Note:** F2 should equal to null, we will correct this typo in the final revision of the paper.
>
> [1] Kiwan Maeng, Chuan Guo, Sanjay Kariyappa, and Edward Suh. Measuring and controlling split layer privacy leakage using fisher information. arXiv preprint arXiv:2209.10119, 2022. [2] Zecheng He, Tianwei Zhang, and Ruby B Lee. Model inversion attacks against collaborative inference. In Proceedings of the 35th Annual Computer Security Applications Conference, pp. 148–162, 2019.

---

> > ### Comment · Reviewer_pjJm · 2025-06-08
> >
> > Thanks the authors for the clarifications which addressed my concerns. The new revision also looks better to me. I'm leaning towards acceptance.

---

### Review · Reviewer_gMLz · 2025-05-22

**Summary Of Contributions:**

This paper investigates feature inversion using deep neural networks, with the goal of reconstructing input images from intermediate representations, such as outputs from hidden layers. Rather than directly optimizing the image pixels, the authors propose optimizing the input latent vector of a latent diffusion model, which is then passed through the reverse diffusion process. The key advantage of this approach is that it leverages the generative power of diffusion models to enhance the quality of the reconstructed images.

The paper considers two scenarios: a white-box setting, where the adversary has full access to the target model, and a black-box setting, where the adversary only observes input-output pairs.

Extensive experiments are conducted across various target models to evaluate the proposed method. It is compared against two baselines: direct pixel optimization and a decoder-based approach. The results show that the proposed method clearly outperforms both baselines. Additional experiments demonstrate that incorporating textual prompts can further enhance reconstruction quality.

**Audience:**

Yes

**Claims And Evidence:**

Yes

**Requested Changes:**

a. The experimental results, especially the quantitative ones, show that using outputs from shallow layers clearly outperforms using outputs from deeper layers. Please elaborate more on whether it is realistic to obtain outputs from shallow layers in practice, especially under the split DNN computing scenario mentioned in the paper.

b. Using textual prompts can improve the quality of the reconstructed images. The authors should discuss more about how these textual prompts can be obtained in practical settings.

c. Although the authors mention that their approach can also be applied to the scenario of end-to-end feature inversion, they do not compare their method with other privacy attack baselines. Meanwhile, more qualitative results should be provided to make the experiments more convincing.

Typo: five lines under Section 3.3

**Strengths And Weaknesses:**

Strengths:

a. The idea of leveraging a powerful diffusion model to recover high-quality input images is interesting and reasonable. The authors use a diffusion model pretrained on a large-scale dataset, which provides strong prior knowledge to support image reconstruction.

b. Both white-box and black-box settings are considered. The inclusion of the black-box scenario adds practical relevance and broadens the applicability of the proposed method.

c. The authors conduct extensive experiments across different target models and various intermediate layers. The results clearly demonstrate that the proposed method outperforms the baselines.

Weaknesses:

a. The novelty of optimizing the latent vector of a generative model is somewhat limited, as this has been explored in prior inversion works. The main contribution lies in adopting a diffusion model rather than other generative models.

b. Beyond pixel-level evaluation metrics, it would be valuable to consider semantic-level metrics that better reflect perceptual quality or semantic similarity.

c. More qualitative results should be provided to strengthen the empirical evidence and improve the overall persuasiveness of the experimental findings.

---

> ### Author Response · Authors · 2025-06-03
>
> Thank you for the insightful review and feedback.
>
> **Q1:** The novelty of optimizing the latent vector of a generative model is somewhat limited, as this has been explored in prior inversion works. The main contribution lies in adopting a diffusion model rather than other generative models.
>
> **A:**  In this work, we demonstrate that diffusion models can significantly enhance reconstruction quality in both white-box and black-box settings. More importantly, we demonstrate that additional information in another format (i.e., textual format) can be utilized to enhance reconstruction performance. This is made possible by the multimodal capabilities of diffusion models, which can integrate additional textual input to generate corresponding images. Leveraging multimodal input for feature inversion is a novel concept that has not been explored in previous research, making our work the first to demonstrate this capability.
>
> **Q2:** Beyond pixel-level evaluation metrics, it would be valuable to consider semantic-level metrics that better reflect perceptual quality or semantic similarity.
>
> **A:** Thanks for the comment, following your comment, we have evaluation our white-box inversion results, specifically, we send the reconstructed image and the original image to the BLIP-v2 model to perform image captioning, and evaluate the textual similarity score of the caption generated using the original image and that with the reconstructed image. Table below shows the embedding-based similarity by sending them to BERT:
>
> | **White-box** | L12 | L24 | L36|
> | --- | --- | --- | --- |
> | DO | 0.93 | 0.77 | 0.41 |
> | DB | 0.95 | 0.84 | 0.66 |
> | DMB | 0.98 | 0.92 | 0.85 |
>
> **Q3:** Please elaborate more on whether it is realistic to obtain outputs from shallow layers in practice, especially under the split DNN computing scenario mentioned in the paper.
>
> **A:** Yes, obtaining outputs from shallow layers is realistic in practice, especially under the Split DNN computing scenario. In such distributed inference setups, early layers are executed on the edge device, and intermediate features are sent to the cloud. If these features are transmitted without privacy-preserving encoding, an adversary can exploit them. In the black-box setting, attackers can reconstruct the original input using only the intermediate results and a trained inversion model. In the white-box setting, if the victim’s model weights are known, reconstruction is even more accurate. The paper shows that shallow-layer features retain more input detail, making them particularly vulnerable to such inversion attacks.
>
> **Q4:**  Using textual prompts can improve the quality of the reconstructed images. The authors should discuss more about how these textual prompts can be obtained in practical settings.
>
> **A:** In split computing scenarios, attackers can often obtain useful prior knowledge to craft effective textual prompts that enhance feature inversion. For instance, in smart home or AR/VR systems, attackers may infer common indoor scenes like a living room or kitchen. In workplace applications like Codec Avatars, they may know the user is in a virtual meeting. Location-based AR apps can leak context through GPS, enabling prompts like “Eiffel Tower on a clear day.” Similarly, in autonomous vehicles or fitness applications, route data or app usage may reveal likely scenes such as “a suburban street” or “a person doing yoga indoors.” These contextual cues, derived from metadata, usage patterns, or known environments, make textual prompt-based inversion attacks more realistic and effective.

---

> > ### Author Response · Authors · 2025-06-03
> >
> > **Q5:**  Although the authors mention that their approach can also be applied to the scenario of end-to-end feature inversion, they do not compare their method with other privacy attack baselines. Meanwhile, more qualitative results should be provided to make the experiments more convincing.
> >
> > **A:** We evaluated our method against DO and DB by replacing the diffusion model in our framework, as detailed in Sections 5 and 6. These baseline methods, DO and DB, were originally introduced in prior works [1][2]. To further assess the generality of our framework, we conducted an additional experiment in which the diffusion model was substituted with a GAN-based generator (StyleGAN3), using a similar pipeline. Specifically, we adapted the workflows shown in Figures 4 and 9 by incorporating StyleGAN3 and optimized its latent vector to reconstruct the user input over ResNet-18. All experiments were conducted under both white-box and black-box settings, maintaining the same training configurations described in Sections 6.1 and 7.1. The quantitative results are summarized in the two tables below, corresponding to the white-box and black-box evaluations, respectively. The results demonstrate that our diffusion-based approach consistently outperforms the GAN-based alternative. We will provide more qualitative results in the final version of the paper.
> > These are the results for the white-box inversion attack:
> > |  | Metric | Method         | L4 | L8   | L12  |
> > |--------|--------|----------------|--------------|------|------|
> > |        | IS     | StyleGAN-based | 7.02         | 6.22 | 5.06 |
> > |        |        | DMB            | 7.23         | 6.86 | 6.48 |
> > |        | PSNR   | StyleGAN-based | 37.3         | 34.1 | 22.5 |
> > |        |        | DMB            | 41.0         | 36.3 | 29.1 |
> > |        | SSIM   | StyleGAN-based | 0.94         | 0.92 | 0.78 |
> > |        |        | DMB            | 0.97         | 0.94 | 0.86 |
> >
> > These are the results for the black-box inversion attack:
> > |        | Metric | Method         | ResNet-18 L4 | L8   | L12  |
> > |--------|--------|----------------|--------------|------|------|
> > |        | IS     | StyleGAN-based | 6.70         | 5.29 | 4.04 |
> > |        |        | DMB            | 6.99         | 6.41 | 5.19 |
> > |        | PSNR   | StyleGAN-based | 36.8         | 27.6 | 11.4 |
> > |        |        | DMB            | 40.4         | 32.5 | 20.6 |
> > |        | SSIM   | StyleGAN-based | 0.92         | 0.80 | 0.41 |
> > |        |        | DMB            | 0.92         | 0.84 | 0.46 |
> >
> > We can see that the results for StyleGAN-based method is not as good as ours.
> >
> > [1] Kiwan Maeng, Chuan Guo, Sanjay Kariyappa, and Edward Suh. Measuring and controlling split layer privacy leakage using fisher information. arXiv preprint arXiv:2209.10119, 2022.
> > [2] Zecheng He, Tianwei Zhang, and Ruby B Lee. Model inversion attacks against collaborative inference. In Proceedings of the 35th Annual Computer Security Applications Conference, pp. 148–162, 2019.

---

### Decision · Action_Editor_vJN5 · 2025-06-27

**Recommendation:** Accept as is

**Audience:**

Yes

**Audience Explanation:**

This paper addresses a critical and timely problem of privacy leakage in deep neural networks, a topic of significant interest to the machine learning community. The main contributions of this work include improving feature inversion to inspect privacy leakage using diffusion models, and extensions that add support for textual priors and inverting features for videos which are interesting to the ML community.

The reviewers all agreed the paper was a good fit for the TMLR audience.

**Claims And Evidence:**

Yes

**Claims Explanation:**

Initially, the reviewers raised valid concerns regarding the novelty, the choice of baselines, the practical applicability of the attack scenarios, and requested additional ablation studies and qualitative results. The authors were responsive and added new experiments comparing their method against a StyleGAN baseline, included a new ablation study on the components of their loss function, and provided more detailed justifications for their experimental setup and the practical relevance of their findings.

The final manuscript presents clear, convincing evidence that diffusion priors enhance image reconstruction quality in feature inversion tasks. All three reviewers unanimously agreed that the revised paper's claims are well-supported by the evidence provided.